# NKG2D upregulation sensitizes tumors to combined anti-PD1 and anti-VEGF therapy and prevents hearing loss

Simeng Lu[1,10], Zhenzhen Yin[1,10], Limeng Wu[1,7], Yao Sun[1,8], Jie Chen[1,9], Lai Man Natalie Wu[2], Janet L. Oblinger[3], Day C. Blake[1], Bingyu Xiu[1], Lukas D. Landegger[4], Richard Seist[4], William Ho[1], Adam P. Jones[1], Alona Muzikansky[5], Konstantina M. Stankovic[4], Scott R. Plotkin[6], Long-Sheng Chang[3] & Lei Xu[1]✉

*NF2*-related schwannomatosis (*NF2*-SWN) is a debilitating condition, characterized by bilateral vestibular schwannomas (VSs) that progressively cause irreversible sensorineural hearing loss. Current management relies on surgery or radiotherapy, while bevacizumab (αVEGF) is used off-label, with variable and often transient efficacy. Effective therapies that durably suppress tumor growth and preserve hearing are urgently needed. Although immune checkpoint inhibitors have transformed cancer treatment, their efficacy in non-malignant tumors such as VS remains unclear. Here, we evaluate combined anti-PD1 (αPD1) and αVEGF therapy in two syngeneic, immune-competent VS models. Combination treatment significantly outperforms either monotherapy, inhibiting tumor growth and preventing hearing loss. Mechanistically, αVEGF enhances αPD1 efficacy by normalizing tumor vasculature, improving drug delivery and immune cell infiltration, and promoting cytotoxicity of T and NK cells via NKG2D upregulation. Combined treatment effectively controls tumor growth that progresses despite anti-VEGF therapy. These findings support αPD1 and αVEGF combination therapy as a promising strategy for *NF2*-SWN.

*NF2*-related schwannomatosis (*NF2*-SWN) is a hereditary neoplastic syndrome resulting from germline pathogenic variants in the *NF2* tumor suppressor gene. It has an incidence rate of about 1 in 61,000 individuals, with nearly complete penetrance at close to 100%[1,2].

Patients with *NF2*-SWN are characterized by bilateral vestibular schwannomas (VS) that arise from cranial nerve VIII. The progressive growth of VSs can lead to sensorineural hearing loss (SNHL), significant morbidity, and greatly affect the quality of life of patients[3]. In some

[1]Edwin L. Steele Laboratories, Department of Radiation Oncology, Massachusetts General Hospital, Harvard Medical School, Boston, MA, USA. [2]Medpace Inc., Cincinnati, OH, USA. [3]Center for Childhood Cancer, Abigail Wexner Research Institute at Nationwide Children's Hospital and Department of Pediatrics, The Ohio State University, Columbus, OH, USA. [4]Department of Otolaryngology – Head and Neck Surgery and Department of Neurosurgery, Stanford University School of Medicine, Stanford, CA, USA. [5]Biostatistics Center, Massachusetts General Hospital, Harvard Medical School, Boston, MA, USA. [6]Department of Neurology and Cancer Center, Massachusetts General Hospital, Harvard Medical School, Boston, MA, USA. [7]Present address: Department of Stomatology, Peking Union Medical College Hospital, Beijing, China. [8]Present address: Department of Radiation Oncology, Tianjin Medical University Cancer Institute and Hospital, National Clinical Research Center for Cancer, Tianjin Clinical Research for Cancer, Key Laboratory of Cancer Prevention and Therapy, Tianjin, China. [9]Present address: Department of Oral and Maxillofacial Surgery, Xiangya Hospital, Central South University, Changsha, Hunan, China. [10]These authors contributed equally: Simeng Lu, Zhenzhen Yin. ✉e-mail: lexu@mgh.harvard.edu

cases, large VSs can put pressure on the brainstem, leading to severe complications and even fatality[4]. Current treatments for progressive VSs include surgery and radiotherapy, both of which carry the risk of further nerve damage and may result in profound deafness, chronic dizziness, and paralysis of the facial and other cranial nerves[5–7]. Bevacizumab, a humanized monoclonal antibody that neutralizes vascular endothelial growth factor-A (VEGF-A), is approved in the UK for the treatment of NF2-SWN and has documented benefits in 30-40% of patients, with improvement in hearing or tumor shrinkage[7–9]. However, not all patients respond to bevacizumab and the effect is not always durable. There remains a significant unmet medical need for effective treatments that can halt VS growth and prevent SNHL associated with these tumors.

Immune checkpoint inhibitors (ICI), including monoclonal antibodies targeting programmed cell death protein-1 (PD-1) and its ligand 1 (PD-L1), work by blocking immune inhibitory signals, thereby enhancing T cell activation and function. This process leads to improved antitumoral immunity and better prognostic outcomes[10,11]. FDA has approved ICIs for a variety of cancers, either as standalone treatments or in conjunction with other therapies such as chemotherapy[12]. However, their therapeutic potential in non-malignant tumors, such as schwannomas, has yet to be investigated.

VS tumors contain a significant population of T cells[13]. However, many of these infiltrating CD4+ and CD8+ T cells express PD-1[14] and display a transcriptome signature indicative of CD8+ T cell senescence[15]. Additionally, PD-1 and its ligand PD-L1 are expressed in VS tumors[13,16]. Despite these findings, there has been limited research on the therapeutic potential of ICIs in VS. Notably, an αPD1 antibody was shown to modestly slow subcutaneous mouse schwannoma growth[17], and αPD1 salvage therapy resulted in tumor growth inhibition in a VS patient with recurrent tumors[18]. These insights indicate that ICIs could represent a promising treatment approach for VS.

In this study, we aim to address two questions: 1) Can αVEGF-induced vessel normalization enhance intratumoral delivery of αPD1 drug and immune effector cells, thus augmenting the ICI efficacy in VS? and 2) Can αPD1 serve as an effective alternative for patients unresponsive to or unable to tolerate bevacizumab? Using schwannoma mouse models, we achieve several key findings: i) we conduct the first comprehensive investigation of the immune checkpoint inhibitor αPD1 as a potential treatment for non-malignant VS, characterizing its effects on tumor growth; (ii) we evaluate the impact of αPD1 on hearing preservation, which has not been previously examined. Importantly, we show that combining αVEGF with αPD1 significantly enhances the efficacy of either monotherapy. Specifically, iii) αVEGF improves drug delivery and promotes intratumoral infiltration of immune effector cells, thereby enhancing αPD1 efficacy; and iv) combined αPD1 with αVEGF treatment effectively controls tumor growth that progresses despite αVEGF treatment. Our long-term goal is to develop therapies for NF2-SWN that both suppress VS tumor growth and preserve hearing. These findings provide a strong foundation for αPD1 with αVEGF combination therapy for patients with NF2-SWN.

## Results

### αVEGF treatment normalizes tumor vasculature and increases αPD1 drug delivery and intratumoral infiltration of immune effector cells in the mouse schwannoma model

To address our first question of whether αVEGF-induced vessel normalization enhances the intratumoral delivery of ICI drugs and immune effector cells, thereby augmenting ICI efficacy in VS, we evaluated how αVEGF treatment affects tumor vasculature in the cerebellopontine angle (CPA) Nf2−/− schwannoma model. αVEGF treatment did not significantly reduce microvessel density (MVD; Fig. 1A-B), but it did significantly increase the fraction of pericyte-covered vessels, indicating that the treated tumor blood vessels are structurally similar to normal vessels (Fig. 1C). Next, to determine whether the structural normalization of tumor vessels leads to normalized vessel perfusion, we measured the fraction of perfused vessels (Fig. 1D). In concert with the blood vessel structural normalization, αVEGF treatment increased the percentage of perfused vessels (Fig. 1E). These data indicate that αVEGF treatment normalized the schwannoma vasculature.

To determine whether anti-VEGF-mediated vascular normalization enhances αPD1 antibody delivery, the αPD1 antibody was conjugated with FITC and administered i.p. to control IgG- or αVEGF-treated mice. Twenty-four hours later, tumor-bearing brains were collected, and the distribution of fluorescently labeled αPD1 antibody was assessed by confocal microscopy (Fig. 1F). We observed significant angiogenesis (CD31+, cyan) at the peritumoral margin, and the FITC-labeled αPD1 antibody (magenta) predominantly accumulated within the tumor, with minimal signal in the adjacent brain tissue (Fig. 1G). In αVEGF-treated tumors, intratumoral green fluorescence intensity was significantly higher than in control IgG-treated tumors (Fig. 1H). Image analysis confirmed that antibody delivery improved following αVEGF treatment (Fig. 1I).

To determine whether αVEGF-mediated vascular normalization increases the intratumoral infiltration of immune effector cells, we analyze the intratumoral immune cell population using flow cytometry (Supplementary Fig. 1). We found that αVEGF treatment significantly increased the number of tumor-infiltrating cytotoxic CD8+ T cells and NK cells, as well as increased the ratio of CD8+ T cells over immune suppressive $T_{reg}$ (CD8+/$T_{reg}$)(Fig. 1J).

### Combined αVEGF treatment enhances αPD1 efficacy in the mouse schwannoma models

The αVEGF-induced increased delivery of αPD1 antibody and immune effector cells provides the rationale to test if αVEGF treatment could enhance αPD1 efficacy. We treated mice bearing Nf2−/− schwannomas in both the sciatic nerve and CPA models with: i) control IgG, ii) αPD1, iii) αVEGF, or iv) αPD1 + αVEGF (Fig. 2A). In both models, all treatments - including monotherapies and the combination therapy - were well tolerated, with no significant body weight loss observed in any treatment group (Supplementary Fig. 2A). In the sciatic nerve model, combined αVEGF treatment enhanced the antitumor effect of αPD1 and significantly delayed tumor growth compared to αPD1 or αVEGF monotherapies (Fig. 2B). This enhanced efficacy was also evident in the SC4 schwannoma model (Supplementary Fig. 2B). In the CPA Nf2−/− schwannoma model, combined αVEGF treatment significantly prolonged animal survival compared to either monotherapy – with 65% of mice surviving ≥100 days in the combination group, compared to 20% of long-survivor mice in the αPD1 group and 25% of long-survivor mice in the αVEGF group (Fig. 2C).

In a separate cohort of Nf2−/− sciatic nerve tumor-bearing mice, all treatments were discontinued after administering three doses. Remarkably, tumor growth remained suppressed even after treatment cessation, indicating a durable antitumor effect that persisted beyond the active treatment phase (Supplementary Fig. 2C-2D).

To examine the effects of immunotherapy on hearing function, we first evaluated potential ototoxicity from αPD1 treatment in non-tumor-bearing mice by measuring ABR, which represents the summed activity of the auditory nerve and central auditory nuclei. In non-tumor-bearing mice, three doses of αPD1 treatment did not alter the ABR threshold in mice 21 days post-treatment (Supplementary Fig. 2E−2F), demonstrating that αPD1 does not cause acute ototoxicity in mice. In mice bearing Nf2−/− tumors in the CPA region, αVEGF treatment prevented tumor-induced hearing loss (blue line), reproducing the hearing benefit from bevacizumab treatment observed in patients with NF2-SWN. αPD1 monotherapy (red line) and its combination with αVEGF treatment (pink line), restored the ABR threshold to normal levels observed in non-tumor-bearing mice. No treatment demonstrated superiority over the others (Fig. 2D).

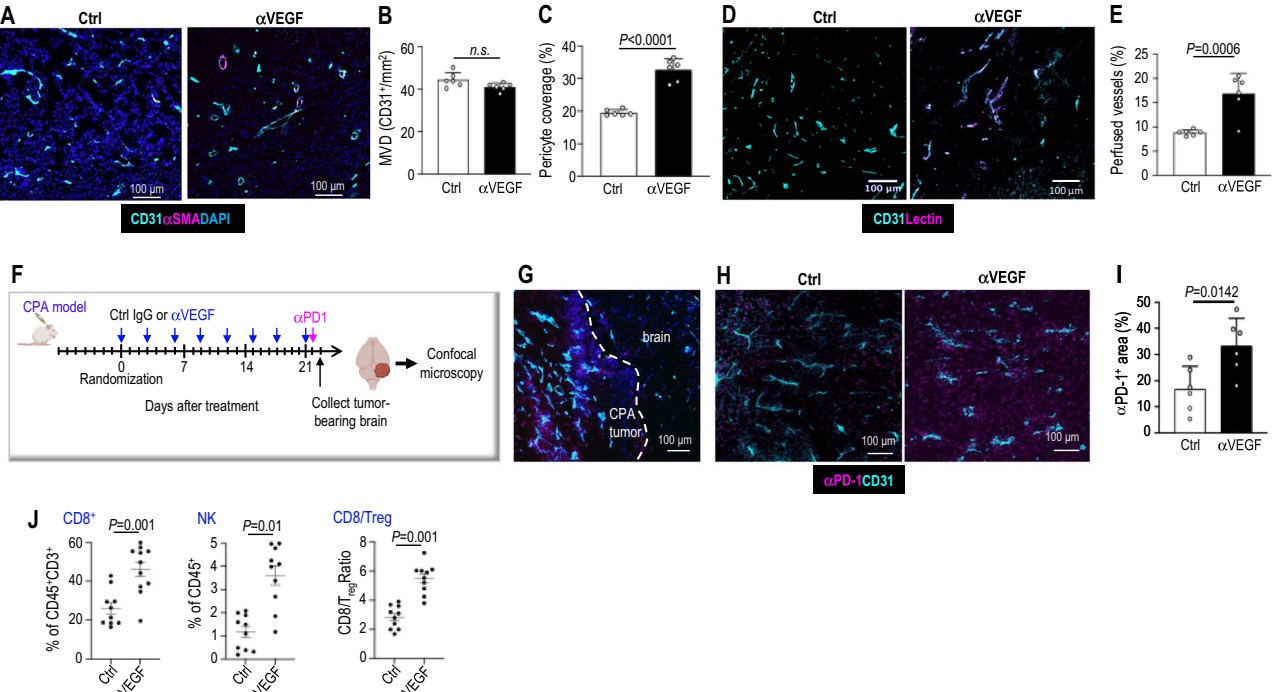

**Fig. 1 | αVEGF treatment normalizes tumor vasculature, increases drug delivery to the tumor, and enhances intratumoral infiltration of immune effector cells in the mouse schwannoma model. A** Representative immunofluorescent staining image of blood vessel endothelial cells (CD31⁺, Cyan) and pericyte (αSMA⁺, magenta) in *Nf2⁻/⁻* tumor treated with control-IgG and αVEGF. DAPI, blue. **B** MVD was quantified by manually counting CD31⁺ (cyan) vessels/mm². *n.s.* not significant. **C** The percentage of pericyte-covered blood vessels (αSMA⁺CD31⁺, purple) of total blood vessels (CD31⁺, cyan) was quantified using ImageJ software. *P* = 0.00008. **D** Representative IF staining images of perfused vessels in *Nf2⁻/⁻* tumor treated with control-IgG and αVEGF. We injected FITC-lectin (magenta) *i.v.* to identify perfused tumor vessels (FITC-lectin⁺/CD31⁺, magenta and purple) and stained for CD31⁺ (cyan) to detect the total number of blood vessels. **E** The percentage of perfused vessels ((FITC-lectin⁺/CD31⁺, magenta and purple)/total blood vessels (CD31⁺, cyan) was quantified using ImageJ. **F-I** Investigate the effects of αVEGF treatment on αPD1 drug delivery. **F** We treated mice bearing *Nf2⁻/⁻* schwannomas in the CPA with either control IgG or αVEGF for 3 weeks. FITC-conjugated αPD1 antibody (cyan) was injected *i.p.* Twenty-four hours later, tumor-bearing brains were harvested for vascular staining and imaged by confocal microscopy to visualize the distribution of fluorescently labeled αPD1 antibody. The schematic in panel F was created in BioRender. Xu, L. (2026) http://BioRender.com/yltfaz6. **G** Representative images of the distribution of FITC-labeled αPD1 antibody in the peritumoral brain and the CPA tumors. Magenta, αPD1 antibody, Cyan, CD31⁺ tumor blood vessels, Blue, DAPI. **H** Representative images of the distribution of FITC-labeled αPD1 antibody (magenta) in control-IgG and αVEGF-treated tumors. Cyan, CD31⁺ tumor blood vessels. **I** Quantification of the fraction of tumor area positive for αPD1 antibody was performed using ImageJ. **J** Flow cytometry quantification of CD8⁺ T cells, NK cells, and CD8/T_reg ratio in *Nf2⁻/⁻* tumors treated with control-IgG (*n* = 10) and αVEGF (*n* = 11). For histological staining, samples from *n* = 6 mice/group were used, 20 random areas/block were imaged and quantified. Image quantification and flow cytometry data are presented as mean ± SD and analyzed using two-sided Student's t-test and the Mann-Whitney test. Source data are provided as a Source Data file.

## αVEGF-enhanced αPD1 efficacy is mediated by CD8⁺ T cells and NK cells

To investigate the mechanisms of how αVEGF enhanced αPD1 efficacy, we first examined the CPA tumor proliferation and apoptosis. In the combination group, we observed fewer proliferating tumor cells (PCNA⁺) and more apoptotic tumor cells (TUNEL⁺) compared to those in the control or monotherapy groups (Supplementary Fig. 3A-3B).

Next, using flow cytometry to profile intratumoral immune cell populations, we found that both αPD1 and αVEGF monotherapies significantly increased the number of tumor-infiltrating cytotoxic CD8⁺ T and NK cells, reduced the number of immunosuppressive MDSCs, compared with the control group. Compared with αPD1 monotherapy, the combination of αVEGF and αPD1 further enhanced the infiltration of effector CD8⁺ T cells and NK cells, while reducing MDSCs (Fig. 3A). Tumor secretome analysis revealed that anti-PD1 treatment elevated the levels of IFN-γ and TNF-α - cytokines known to synergistically activate T cells and produced by activated T cells. Combined αVEGF and αPD1 treatment further increased IFN-γ and TNF-α production compared with αPD1 monotherapy (Fig. 3B). These findings suggest that, beyond recruiting more CD8⁺ T and NK cells to the tumor, αVEGF treatment may also enhance the activation of these immune effector cells. Consistent with this, flow cytometry analysis showed that αVEGF treatment significantly increased the proportion of CD8⁺ T cells and NK cells expressing granzyme B and perforin - key mediators of cytotoxicity in these effector cells (Fig. 3C and Supplementary Fig. 4).

To investigate if αVEGF-driven infiltration and activation of CD8⁺ T cells and NK cells are essential for the enhanced antitumor efficacy of the combination therapy, we depleted mice of CD8⁺ T cells or NK cells prior to treatment (Fig. 3D). αCD8 antibodies effectively deplete CD8⁺ T cells, and αNK1.1 antibodies specifically reduce NK cells, as confirmed by flow cytometry (Fig. 3E). CD8⁺ T cell depletion (pink line) completely abolished the therapeutic benefit of the combination therapy, whereas NK cell depletion (blue line) significantly reduced the antitumor response (Fig. 3F).

## NKG2D upregulation enhances T and NK cell cytotoxicity, sensitizes tumors to combined αPD1 and αVEGF therapy

To investigate whether αVEGF treatment regulates the cytotoxic function of NK cells and CD8⁺ T cells, from control- and αVEGF-treated mice, we isolated: i) tumor-associated CD8⁺ T cells, and ii) spleen NK cells, given the low number of tumor-infiltrating NK cells. CD8⁺ T cells were co-cultured with calcein AM-labeled *Nf2⁻/⁻* tumor cells at an effector-to-target ratio (E:T) of 10:1. Similarly, NK cells were co-cultured with calcein AM-labeled *Nf2⁻/⁻* tumor cells or YAC-1 cells,

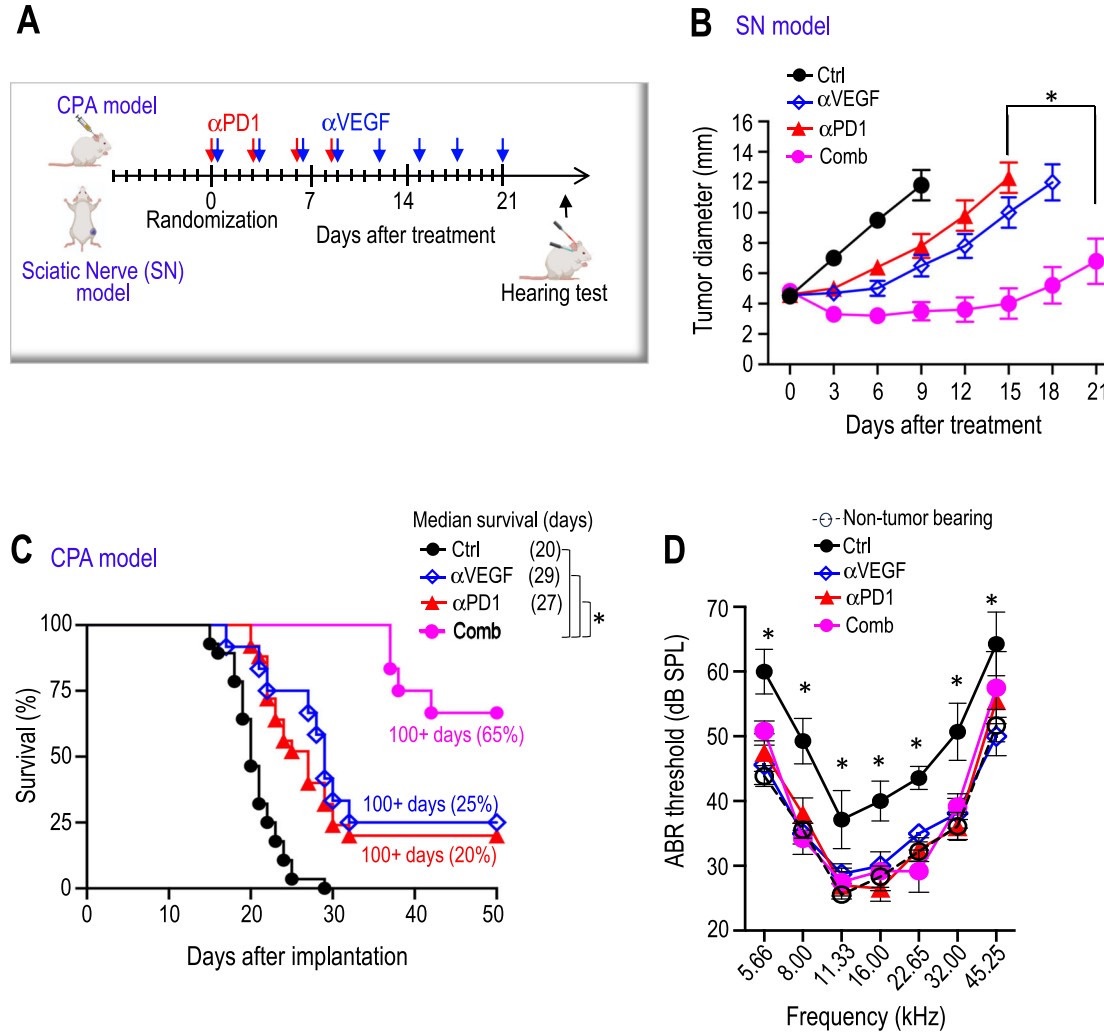

**Fig. 2 | Combined αVEGF treatment enhances αPD1 efficacy in the mouse schwannoma models.** Diagram showing the timeline of αVEGF and αPD1 combination treatment in the CPA and SN mouse schwannoma models. Tumor-bearing mice were treated with control IgG, αPD1 (200 μg/mice), αVEGF (2.5 mg/kg), or combined αPD1 and αVEGF. The schematic in **A** was created in BioRender. Xu, L. (2026) http://BioRender.com/yltfaz6. **B** In the sciatic nerve model, tumor size was measured by caliper every 3 days post-treatment. *$P = 0.00007$. $N = 18$ mice/group. **C** In the CPA model, animal survival is recorded by the Kaplan-Meier survival curve. *$P = 0.00006$. **D** A hearing function test was performed on the CPA model when mice exhibited ataxia. ABR threshold as a function of frequency in mice treated with control IgG (Ctrl), αPD1, αVEGF, and combination antibodies (Comb). Non-tumor-bearing mice served as a baseline control. Ctrl *vs.* αPD1: 5.66 kHz: $P = 0.005$,

8.0 kHz: $P = 0.001$, 11.33 kHz: $P = 0.004$, 16.0 kHz: $P = 0.0002$, 22.65 kHz: $P = 0.001$, 32.0 kHz: $P = 0.00005$, 45.25 kHz: $P = 0.013$; Ctrl *vs.* αVEGF: 5.66 kHz: $P = 0.0001$, 8.0 kHz: $P = 0.0001$, 11.33 kHz: $P = 0.024$, 16.0 kHz: $P = 0.007$, 22.65 kHz: $P = 0.02$, 32.0 kHz: $P = 0.008$, 45.25 kHz: $P = 0.0001$; Ctrl *vs.* Comb: 5.66 kHz: $P = 0.022$, 8.0 kHz: $P = 0.002$, 11.33 kHz: $P = 0.016$, 16.0 kHz: $P = 0.006$, 22.65 kHz: $P = 0.0004$, 32.0 kHz: $P = 0.004$, 45.25 kHz: $P = 0.01$. $N = 18$ mice/group. All animal studies are representative of at least three independent experiments, with graphs depicting the mean ± SEM. Differences in sciatic nerve tumor growth were analyzed using repeated-measures two-way ANOVA. Kaplan-Meier survival curves were analyzed by the Log-rank (Mantel-Cox) test. ABR thresholds were analyzed with a linear mixed-effects model.

which are sensitive to NK cell-mediated cytotoxicity, at the same E:T ratio. CD8+ T cells and NK cells isolated from αVEGF-treated mice exhibited significantly enhanced cytotoxicity against both *Nf2*−/− tumor cells and YAC-1 target cells compared to effector cells isolated from control IgG-treated mice (Fig. 3G).

We next investigated whether systematic αVEGF treatment activates CD8+ T cells and NK cells. In *Nf2*−/− tumors, αVEGF treatment significantly increased the expression of natural killer group 2D (*Nkg2d*)(Fig. 3H). NKG2D is an activating receptor expressed on NK cells, CD8+ T cells, and activated CD4+ T cells, which promotes NK cell cytotoxicity[19] and augments T cell receptor-mediated activation of CD8+ T cells[20]. In mice, NKG2D binds to its ligands RAE-1 and H60A[21,22]. αVEGF treatment also significantly increased the expression of *Rae-1* and *H60a*, but did not affect the expression of the inhibitory receptor *Nkg2a* (Fig. 3H).

Lastly, to determine whether NKG2D signaling is required for the enhanced antitumor efficacy of the combination therapy, we blocked NKG2D signaling using a neutralizing antibody. *Nf2*−/− tumor-bearing mice were treated with i) control IgG, ii) αVEGF, iii) αVEGF + αNKG2D neutralizing antibody, iv) αVEGF + αPD1, or v) αVEGF + αPD1 + αNKG2D (Fig. 3I). Blockade of NKG2D signaling markedly attenuated the anti-tumor response to both αVEGF monotherapy and combination treatment (Fig. 3J), indicating that NKG2D signaling is critical for the therapeutic benefit from the αVEGF treatment.

**scRNA-Seq analysis reveals elevated cytotoxic profiles of CD8+ T and NK cells in VS from an *NF2*-SWN patient treated with bevacizumab**

To profile the T cells and NK cells in VSs with or without bevacizumab treatment, we dissociated fresh VS tissues from *NF2*-SWN patients who

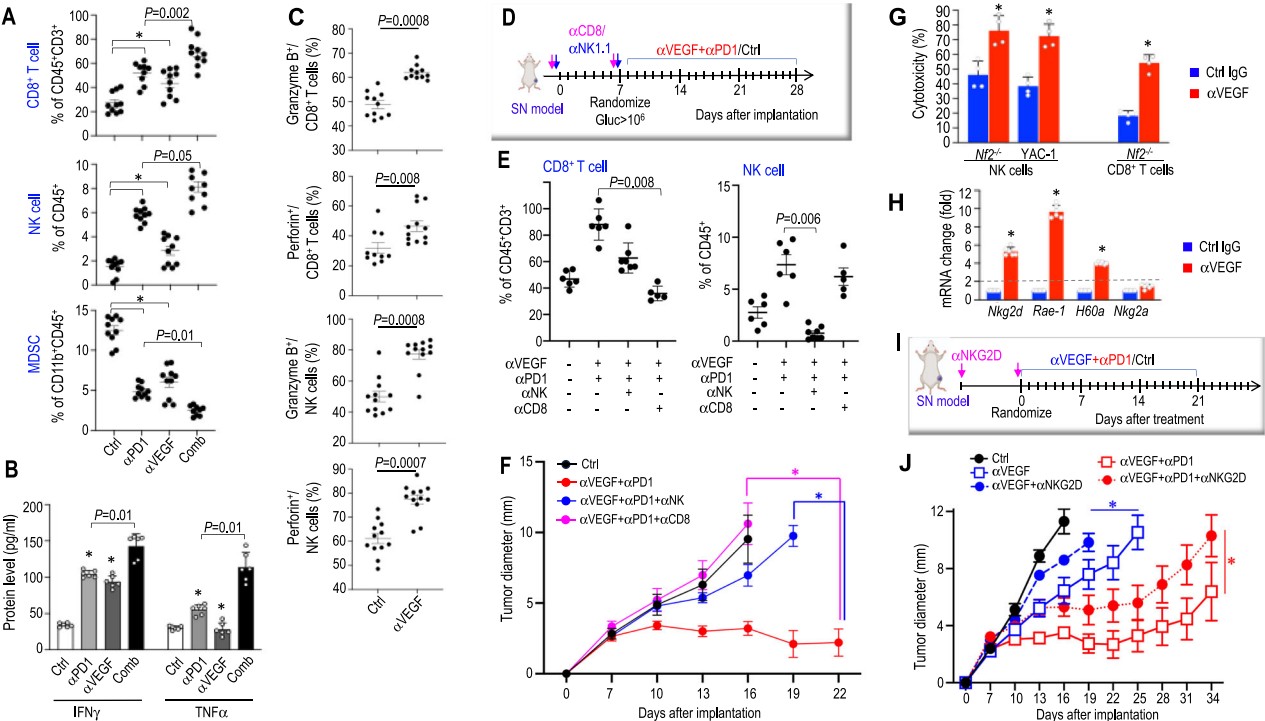

**Fig. 3 | NKG2D upregulation enhances T and NK cell cytotoxicity, sensitizes tumors to combined αPD1 and αVEGF therapy. A** Flow cytometry analysis of the number of CD8⁺ T cells, NK cells, and MDSC in *Nf2⁻/⁻* tumors treated with control IgG (Ctrl, *n* = 10), αPD1 (*n* = 9), αVEGF (*n* = 10), and combination antibodies (Comb, *n* = 10). *\*P* = 0.01. **B** ELISA of murine IFNγ and TNFα in *Nf2⁻/⁻* tumor from different treatment groups (*N* = 6 mice/group). *P* = 0.01. **C** Flow cytometry analysis of the proportion of CD8⁺ T cells expressing granzyme B (*P* = 0.0008) and perforin (*P* = 0.008) and NK cells expressing granzyme B (*P* = 0.0008) and perforin (*P* = 0.0007) in *Nf2⁻/⁻* tumors treated with control IgG (*n* = 11) or αVEGF antibodies (*n* = 12). **D** Schematic and timeline of CD8 T cell and NK cell depletion in *Nf2⁻/⁻* sciatic nerve model. Mice bearing *Nf2⁻/⁻* tumors in the sciatic nerve were treated with anti-CD8 (200 μg/mouse) and anti-NK (200 μg/mouse) antibodies, followed by control IgG, αPD1, αVEGF antibody treatments two days later. **E** Flow cytometry analysis of the percentage of CD8⁺ T cells and NK cells in *Nf2⁻/⁻* tumors treated with control IgG (Ctrl, *n* = 6), αPD1 + αVEGF (*n* = 7), αPD1 + αVEGF + αNK (*n* = 7, *P* = 0.006 compared to αPD1 + αVEGF), and αPD1 + αVEGF + αCD8 (*n* = 5, *P* = 0.008 compared to αPD1 + αVEGF). **F** Sciatic nerve *Nf2⁻/⁻* tumor diameter was measured with a caliper every 3 days following treatment. αPD1 + αVEGF *vs.* αPD1 + αVEGF + αCD8: *P* = 0.001 (pink); αPD1 + αVEGF *vs.* αPD1 + αVEGF + αNK: *P* = 0.003 (blue). *N* = 24 mice/group. **G** In vitro T cell and NK cell cytotoxicity assay. *Nf2⁻/⁻* tumor-bearing mice were treated with control IgG or αVEGF for 21 days. Tumor-associated CD8⁺

T cells and spleen NK cells were isolated. CD8⁺ T cells were stimulated with IL-2 (50 IU/ml). CD8⁺ T cells and NK cells were co-cultured with *Nf2⁻/⁻* tumors and YAC-1 NK target cells at an E:T ratio of 10:1. After 8 hours of co-culture, target cell lysis was quantified by measuring the fluorescent intensity in the supernatant. *P* = 0.0001. *N* = 6 mice/treatment, in vitro cytotoxicity assay performed in triplicates for each condition. **H** qRT-PCR analysis of murine *Nkg2d, Nkg2a, Rae-1,* and *H60a* mRNA in *Nf2⁻/⁻* tumor (*N* = 6 tumors/group). *P* = 0.0001. **I** Schematic and timeline of NKG2D blockade in *Nf2⁻/⁻* sciatic nerve model. Mice bearing *Nf2⁻/⁻* tumors in the sciatic nerve were treated with anti-NKG2D (αNKG2D, 200 μg/mouse, *i.p.*, for two doses: seven days before implantation and seven days after implantation), followed by control IgG, αPD1, αVEGF antibody treatments. The schematic in panel I was created in BioRender. Xu, L. (2026) http://BioRender.com/yltfaz6. **J** Sciatic nerve *Nf2⁻/⁻* tumor diameter was measured with a caliper every 3 days following treatment. αVEGF *vs.* αVEGF + αNKG2D: *P* = 0.02 (red); αVEGF + αPD1 *vs.* αVEGF + αPD1 + αNKG2D: P = 0.04 (blue). *N* = 24 mice/group. Flow cytometry, ELISA, qPCR, and cytotoxicity studies are presented as mean ± SD, and analyzed using two-sided Student's *t* test and the Mann-Whitney U test. All animal studies are presented as mean ± SEM, representative of at least three independent experiments. Differences in tumor growth were analyzed using repeated-measures two-way ANOVA. Source data are provided as a Source Data file.

were either bevacizumab-naïve or treated with bevacizumab for about 10 years. Single-cell transcriptomic profiling was performed using the 10X Genomics platform. Within the VS tissues, cell clusters were partitioned into major cell types, including Schwann cells, macrophages, lymphocytes, fibroblasts, and other stromal cells (Chang LS, et al., manuscript in preparation). We further subclustered the lymphocyte compartment and identified distinct lymphocyte-derived cell states, including CD8⁺ cytotoxic T cells, early naïve T cells, CD8⁺ PD-1⁺ T cells, regulatory T cells, and NK cells in naïve VSs (Fig. 4A).

In NK cells and CD8⁺ T cells from naïve VSs, we observed elevated RNA expression of i) *KLRC1*, which encodes NKG2A, a major inhibitory receptor expressed on NK cells and CD8⁺ T cells[23,24]; ii) *KLRD1*, which encodes CD94, forming a heterodimer with NKG2A; and iii) *HLA-E*, the ligand for NKG2A:CD94 heterodimer, which sends a strong inhibitory signal regulating the cytotoxic activity of NK and CD8⁺ T cells (Fig. 4B)[25,26].

In bevacizumab-treated VS, compared to naïve VS, we observed changes in RNA expression of CD8⁺ T cell and NK cell cytotoxicity

markers: i) reduced expression of inhibitory receptors on NK cells and T cells, including *KIR2DL3, KIR2DL2, KIR3DL1, KIR3DL2*[27,28], ii) reduced expression of immune checkpoint proteins and receptors, including *CD96, PVRIG, HLA-E*[29–31], iii) increased expression of molecules mediating NK cell and CD8⁺ T cell cytotoxicity and recruitment, including Granzyme B (*GZMB*), the key mediator of[32], C-C motif chemokine ligand 4 (*CCL4*), chemokine recruiting NK cells and T cells[33], *CLEC2B and SH2D1B* which activates NK cells[34,35], and iv) NFkB pathway genes, *including REL, NFkB1,* and *NFkB2*, which play a crucial role in regulating the cytotoxic activity of both NK cells and T cells[36] (Fig. 4C-4D and Supplementary Fig. 5). These scRNASeq findings indicate that bevacizumab treatment enhances the cytotoxic function of CD8⁺ T and NK cells.

### αPD1 treatment can control the growth of tumors that progress despite αVEGF treatment in Schwannoma models

We next experimented to address our second question: can αPD1 serve as an effective alternative for patients who are unresponsive to or unable to tolerate bevacizumab? We implanted mouse *Nf2⁻/⁻* cells in

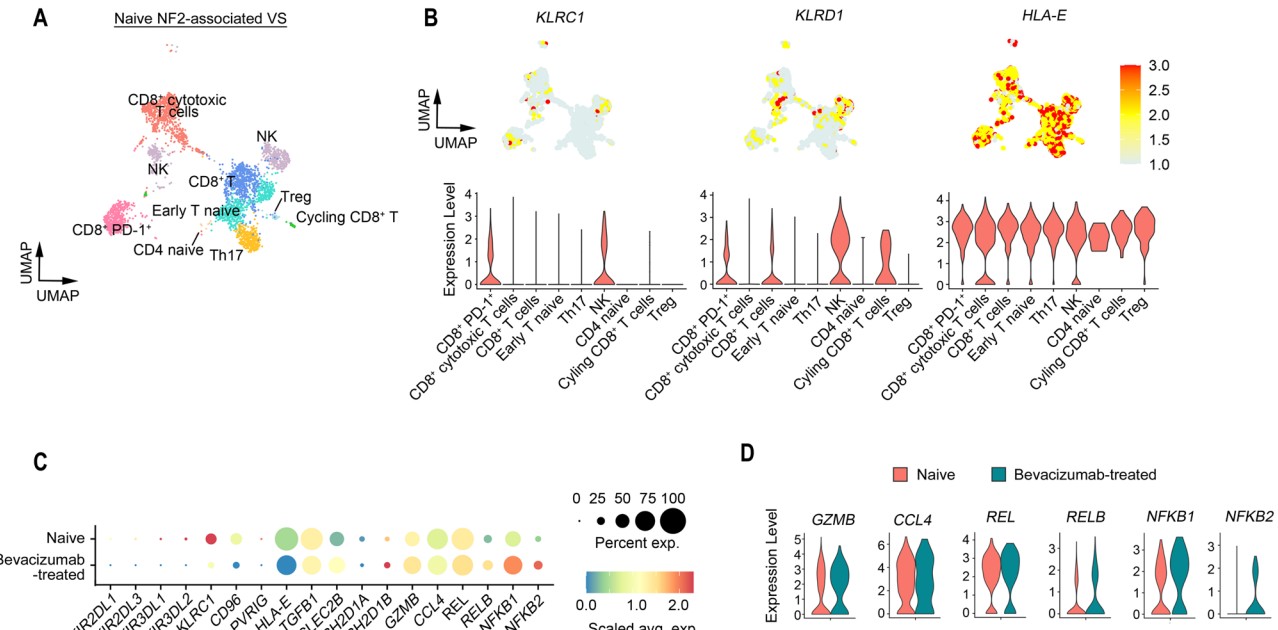

**Fig. 4 | scRNA-Seq reveals elevated cytotoxic profiles of CD8⁺ T and NK cells in VS from an *NF2* patient treated with bevacizumab. A** UMAP visualization of various lymphocyte subpopulations from treatment-naïve *NF2*-associated VS (*n* = 3). Colors represent assigned cell types. **B** The gene expression (top) and violin plots (bottom) of *KLRC1, KLRD1,* and *HLA-E* expression within lymphocyte sub-populations in naïve *NF2*-associated VS (*n* = 3). **C** The dot plot illustrates the expression levels of activating or inhibitory receptors, ligands, and cytokines in NK cells in the VS with (*n* = 1) or without (*n* = 3) bevacizumab treatment. The color scale represents the scaled average expression of the indicated marker genes in untreated or treated VS, and the size of the circle indicates the proportion of cells expressing each marker gene. **D** The violin plots of cytotoxicity marker gene expression in NK cells in the VS with (*n* = 1) or without (*n* = 3) bevacizumab treatment.

both the sciatic nerve and CPA models and treated the mice with three doses of αVEGF. Then, the mice were randomized into three groups, receiving: i) continued αVEGF monotherapy, ii) combined αPD1 with αVEGF treatments, or iii) switched to αPD1 monotherapy, with αVEGF discontinued (Fig. 5A).

In the sciatic nerve model, both switching to αPD1 monotherapy and combining αPD1 with αVEGF significantly delayed tumor growth compared to αVEGF alone, with the combination therapy demonstrating the greatest efficacy (Fig. 5B). In the CPA model, while switching to αPD1 modestly prolonged animal survival, discontinuing αVEGF completely abolished the hearing benefit (red line). Combining αPD1 with αVEGF treatment led to the most significant survival benefit and further prevented tumor-induced hearing loss compared to αVEGF alone (pink line)(Fig. 5C, D).

Flow cytometry analysis revealed that both switching to αPD1 and combining αPD1 with αVEGF significantly increased intratumoral CD8⁺ T cells and NK cells while reducing immune suppressive MDSCs compared to αVEGF monotherapy (Fig. 5E). These findings suggest that αPD1 antibody can serve as an effective alternative or be combined with αVEGF in treating VS. Furthermore, the αPD1 and αVEGF combination provides superior efficacy in tumor control and hearing preservation.

## Discussion

ICIs have transformed cancer therapy in recent years; however, their application to non-malignant schwannomas and their therapeutic potential on hearing preservation remains unexplored. Our study filled this gap, making several discoveries: First, we characterized the effects of αPD1 in various *Nf2*⁻/⁻ schwannoma models, marking the first comprehensive investigation of an ICI as a potential treatment for non-malignant VS. A previous study using a subcutaneous mouse model reported that αPD1 antibody modestly delayed schwannoma growth[17]. As a step further, we used two anatomically correct models of

schwannoma: i) the CPA model, which reproduces the intracranial microenvironment and recapitulates tumor-induced hearing loss, and ii) the sciatic nerve schwannoma model, which reproduces the nerve microenvironment of peripheral nerve schwannomas. Our results suggest that immune checkpoint molecules are valid therapeutic targets for VS, as we showed that αPD1 treatment impeded schwannoma growth. In the clinic, a case report has demonstrated that αPD1 salvage therapy resulted in tumor growth inhibition in a VS patient with recurrent tumors[18]. Our preclinical findings and this case report provide the rationale for future translational studies to better characterize the efficacy of ICIs in patients with VS.

The current study also represents the first investigation of αPD1 therapy on hearing preservation. We investigated potential ototoxicity from αPD1 treatment. Debilitating and sometimes life-threatening immune-related adverse events can result from aberrant activation of T cell responses following ICIs therapy[37]. A growing body of research indicates that inflammation plays a critical role in hearing loss[38,39]. In our mouse model, we observed no acute ototoxic effects from αPD1 treatment and no change in hearing function for up to 3 weeks post-treatment. These findings suggest that αPD-1 may have a manageable safety profile in terms of ototoxicity; however, close monitoring of inflammatory biomarkers in future clinical investigations will be necessary to assess the overall safety of ICI therapy in VS treatment.

The benefits of ICIs are limited to a subset of patients, with efficacy reported in fewer than 20-30% of patients with non-small cell lung cancer, renal cell carcinoma, and melanoma[40]. In these cancers, the density and spatial distribution of immune infiltrates in the tumors are significantly associated with patient survival and response to immune therapy[41–43]. Abnormal vascular perfusion impedes the intratumoral infiltration of immune cells[44,45]. In our VS mouse models, we previously showed that αVEGF normalizes schwannoma vasculature and improves vessel perfusion[46]. This finding prompted us to investigate whether αVEGF-improved vessel perfusion could enhance ICI drug

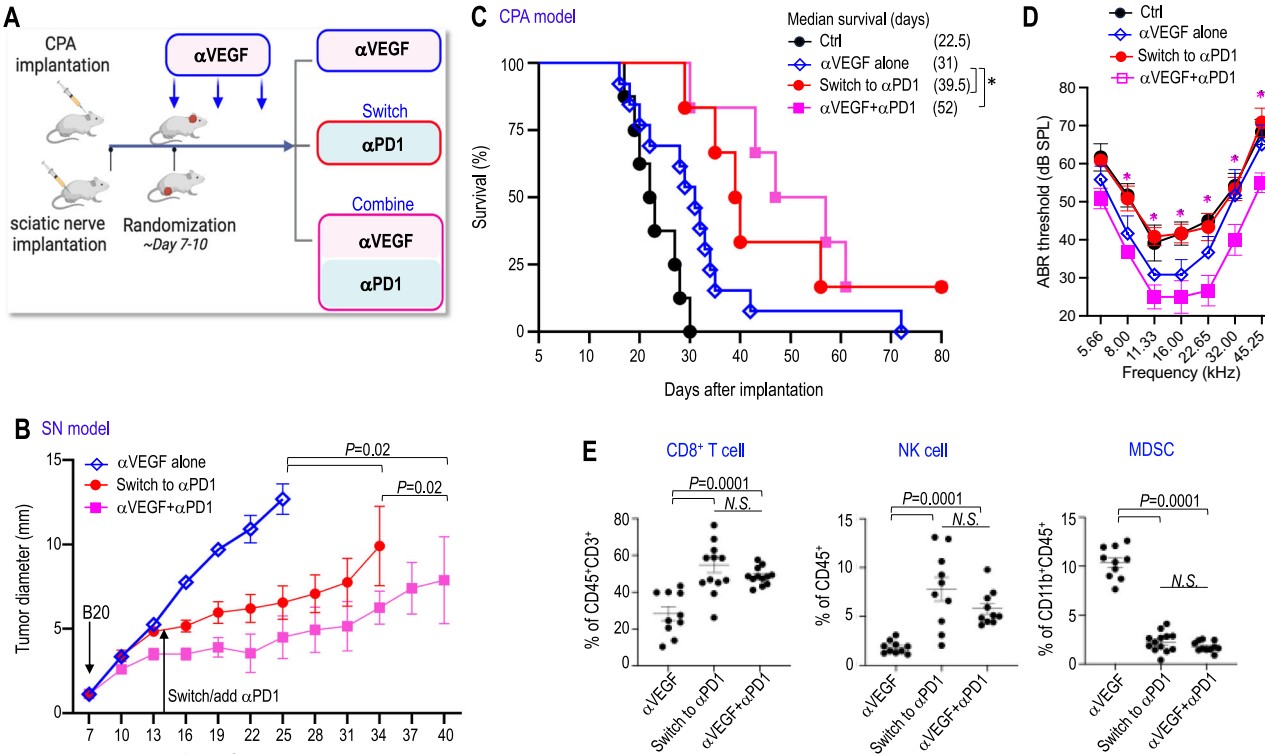

**Fig. 5 | αPD1 can control the growth of tumors that progress despite αVEGF treatment in mouse schwannoma models. A** Graphic depicting the timeline of αVEGF and αPD1 treatment in the *Nf2⁻/⁻* CPA and SN models. The schematic in panel A was created in BioRender. Xu, L. (2026) http://BioRender.com/twbqhmz. **B** In the SN model, tumor diameter was measured by caliper every 3 days. *N* = 24 mice/ group. **C** Kaplan-Meier survival curve of mice bearing *Nf2⁻/⁻* tumor in the CPA model. *\*P* = 0.0001. *N* = 24 mice/group. **D** ABR threshold of mice bearing *Nf2⁻/⁻* tumor in the CPA model. αVEGF + αPD1 *vs.* switch to αPD1: 8.0 kHz: *P* = 0.018, 11.33 kHz: *P* = 0.005, 16.0 kHz: *P* = 0.003, 22.65 kHz: *P* = 0.003, 32.0 kHz: *P* = 0.031, 45.25 kHz: *P* = 0.005. *N* = 18 mice/group. **E** Flow cytometry analysis of the number of CD8⁺ T cells, NK cells, and MDSC in *Nf2⁻/⁻* tumors from αVEGF alone (*n* = 10), Switch to αPD1 (*n* = 10), αVEGF + αPD1 (*n* = 10). Flow cytometry studies are presented as mean ± SD, with significance analyzed by two-sided Student's *t* test and the Mann-Whitney U test. Animal experiments are graphed as mean ± SEM. Graphs are representative examples of at least three independent experiments. Differences in sciatic nerve tumor growth were analyzed using repeated-measures two-way ANOVA. Kaplan-Meier survival curves were analyzed by the Log-rank (Mantel-Cox) test. ABR thresholds were analyzed with a linear mixed-effects model. Source data are provided as a Source Data file.

delivery and immune cell infiltration, thereby augmenting ICI efficacy in VS? Interestingly, we found that αVEGF enhances αPD1 efficacy not only by normalizing the tumor vasculature to improve drug delivery and immune cell infiltration, but also by activating the antitumor cytotoxicity of T cells and NK cells via NKG2D upregulation. The enhanced cytotoxicity of NK cells observed in the spleen following αVEGF treatment likely reflects a systemic immune-modulatory effect of VEGF blockade. VEGF is known to suppress immune cell maturation and cytotoxic function[47], thus its inhibition can restore NK cell activity in peripheral lymphoid organs. Here, we further demonstrated that αVEGF treatment can activate NK cell activity by upregulating the expression of the NK cell activating receptor, NKG2D. The activation of tumor-associated CD8⁺ T cells is more dependent on tumor-specific antigen presentation within the tumor microenvironment. Future studies are needed to further investigate the systemic versus local immune effects of αVEGF treatment.

Bevacizumab, a humanized monoclonal antibody that neutralizes VEGF-A, is approved in the UK for the treatment of *NF2*-SWN and has documented benefits in 30–40% of *NF2*-SWN patients, with improvement in hearing or tumor shrinkage[9]. However, significant challenges remain: i) not all patients respond to bevacizumab, ii) hearing preservation is often not durable in responders, and iii) some patients may be unable to tolerate long-term treatment due to adverse effects[48]. Therefore, there is a need for combination regimens or novel therapies to control VS tumors that progress despite bevacizumab treatment. In our *Nf2⁻/⁻* tumor-bearing mice treated with αVEGF treatment, we

demonstrated that either switching to αPD1 treatment or adding αPD1 to αVEGF treatment more effectively delayed tumor growth and prolonged survival compared with continuing αVEGF monotherapy. Notably, αVEGF treatment was essential for hearing preservation – discontinuing αVEGF and switching to αPD1 alone abolished the hearing protection, whereas combining αPD1 with αVEGF produced the most robust benefit in hearing preservation. The enhanced tumor control observed after switching from αVEGF to αPD1 likely reflects a sustained effect of αVEGF-induced vascular normalization. αVEGF pre-treatment improves tumor vessel perfusion and immune cell infiltration, thereby creating a more immunostimulatory microenvironment that enhances αPD1-mediated cytotoxicity. Together, these findings suggest that i) αPD1 therapy may offer a promising alternative for *NF2*-SWN patients who are refractory to or intolerant of bevacizumab; and ii) combined αVEGF and αPD1 therapy represents a clinically translatable strategy that both sustains vascular normalization and reinvigorates anti-tumor immunity.

In preparation for clinical translation, we performed scRNA-seq on vestibular schwannoma tumors from patients with *NF2*-SWN, with and without long-term bevacizumab treatment. Our analysis confirmed that bevacizumab treatment is associated with increased RNA expression of CD8⁺ T cell and NK cell cytotoxicity markers, supporting an immune-stimulatory effect of αVEGF therapy in patients. A limitation of this analysis is the small number of patient samples, particularly from bevacizumab-treated VSs. Given the rarity of VS, particularly with long-term bevacizumab treatment, and the surgical risks in previously

treated patients, access to such specimens remains limited. Future studies with larger, multi-institutional patient cohorts will be critical to validate and better represent the heterogeneity of patient response.

Clinical trials combining αVEGF therapy with immune checkpoint inhibitors have been conducted across multiple cancers, including renal cell carcinoma (RCC), hepatocellular carcinoma, and non-small cell lung carcinoma. Several studies demonstrated promising results in improving progression-free survival and/or overall survival, for example, pembrolizumab with axitinib in advanced-stage renal cell carcinoma (KEYNOT-426)[49], pembrolizumab with lenvatinib (CLEAR trial)[50,51], and nivolumab plus cabozantinib (CheckMate9ER trial)[52]. These positive outcomes led to regulatory approvals of VEGFR tyrosine kinase inhibitors (TKI) with αPD1 antibody combinations for RCC. However, similar combination strategies have not shown survival benefit in the first-line setting for endometrial cancer (LEAP-001 trial), gastroesophageal cancer (NCT04662710), breast cancer (NCT04732598), and colorectal cancer (NCT02997228)[53]. Our preclinical results provide a strong rationale for clinical translation of immune checkpoint blockade, either as adjunctive therapy or in a sequential setting following bevacizumab, for patients with progressive vestibular schwannomas. Guided by these results and informed by insights from the successful combination trials in other malignancies, a clinical trial (INTUITT) investigating combined αVEGF and αPD1 therapy is currently being planned at Massachusetts General Hospital.

In summary, our study provides the first systematic characterization of the effects of an ICI on non-malignant schwannomas and its potential therapeutic benefit on hearing preservation. Our findings warrant further investigation of the efficacy of αPD1 in combination with standard-of-care treatments for VS. The approach may result in rapid translation to the clinic to improve treatment efficacy and outcomes for patients afflicted by *NF2*-SWN.

## Methods

### Cell lines
Mouse *Nf2*⁻/⁻ Schwann cells were grown in Schwann cell medium containing 10% fetal bovine serum (FBS), which includes Schwann cell growth supplement (SCGS, ScienCell)[54]. Mouse SC4 Schwannoma cells (gift from Dr. Vijaya Ramesh, Massachusetts General Hospital, MGH) were maintained in Dulbecco's Modified Eagle Medium (DMEM, Corning) containing 10% FBS[46,55].

### Animal models
*Nf2*⁻/⁻ and SC4 tumor cells were inoculated into immunocompetent C57/FVB mice. Both male and female mice (1:1 ratio), aged 8-12 weeks, were used to ensure sufficient statistical power and to examine potential sex-related differences. Given that schwannomas develop in patients' vestibular and peripheral nerves, we employed two mouse models by injecting tumor cells into the cerebellopontine angle or into the sciatic nerve.

**Cerebellopontine angle (CPA) model.** To simulate the intracranial microenvironment of VSs, tumor cells were injected into the CPA region of the right hemisphere[56,57]. Each mouse received an implantation of 1 μl of tumor cell suspension containing 2,500 cells.

**Sciatic nerve (SN) schwannoma model.** To mimic the microenvironment associated with peripheral schwannomas, we injected schwannoma cells into the sciatic nerve of the mice[46]. Tumor cell suspension ($5 \times 10^4$ cells in 3 μl) was slowly injected over the course of 45-60 seconds under the sciatic nerve sheath to avoid any leakage.

All mice were housed in the Massachusetts General Hospital (MGH) specific pathogen–free animal facility under a 12 h light/12 h dark cycle, at an ambient temperature of 20–24 °C and relative humidity of 30–70%, with ad libitum access to food and water. All

experiments were conducted in accordance with the approved animal protocol, and no experimental endpoints exceeded those specified in the protocol.

The experimental endpoint for the CPA model is the onset of neurological symptoms in mice. The experimental endpoint for the SN model is when the tumor grows to ~1 cm in diameter. At the experiment endpoint, mice were euthanized by i.p. administration of penobarbital at an overdose dose (150-200 mg/kg).

### Treatment protocols
In the SN model, tumor diameter was measured by caliper, and treatment began once the tumor reached a diameter of 3 mm. In mice with CPA tumors, we started treatment when the blood concentration of the *Gaussia* luciferase (Gluc) reporter gene expression reached $1 \times 10^4$ RLU (relative luminescence unit). For αPD1 treatment, the αPD1 antibody or isotype IgG control (200 μg/mouse, BioXCell) was delivered *i.p.* in 4 doses every 3 days[58]. For αVEGF treatment, the αVEGF antibody B20.4.4 (B20, 2.5 mg/kg, Genentech), which neutralizes both human and mouse VEGF, was administered *i.p.* once every week[46]. Detailed antibody information is provided in Supplementary Table 1.

### Measurement of tumor growth
To track tumor growth in the CPA model, both tumor cell lines were transfected with lentivirus carrying the secreted Gluc reporter gene, and plasma Gluc was measured as previously described[57,59–61]. In brief, 13 μl of whole blood was collected from the tail vein and mixed with 5 μl of 50 mM EDTA to prevent clotting. The blood sample was then transferred to a 96-well plate, where Gluc activity was assessed using a GloMax 96 Microplate Luminometer (Promega). The luminometer was configured to automatically inject 100 μl of 100 mM coelenterazine (Nanolight) in PBS, with photon counts recorded for 10 s. The tumor size in the sciatic nerve was measured with calipers every 3 days until the tumors reached a diameter of 1 cm.

### Audiometric testing in mice
Auditory brainstem responses (ABRs) were assessed as previously outlined[57]. In brief, the animals were anesthetized by intraperitoneal injection of ketamine (0.1 mg/g) and xylazine (0.02 mg/g). All mice were verified by microscopic examination of the tympanic membrane and the middle ear to have well-aerated middle ears and no signs of otitis media, and all animals showed. ABRs were collected using subdermal needle electrodes positioned with the positive electrode placed on the inferior aspect of the ipsilateral pinna, the negative electrode at the vertex, and the ground electrode at the base of the tail. The responses were amplified 10,000-fold, filtered within the 0.3–3.0 kHz range, and averaged over 512 repetitions for frequency and sound levels. Data acquisition was achieved using custom LabVIEW software operating on a PXI chassis from National Instruments Corp. For each frequency tested, the auditory threshold was determined as the lowest stimulus level at which consistent peaks could be visually identified. If no auditory threshold was detected, a value of 85 dB was recorded, representing 5 dB above the highest level tested.

### Flow cytometry analysis of tumor-infiltrating immune cells
To analyze the immune cells that infiltrate the tumor, tumor lysates were dissociated into single cells in digestion buffer (collagenase P 0.2 mg/ml, dispase II 0.8 mg/ml, DNase 0.1 mg/ml, RPMI with 1% FBS) at 37 °C. Every 5-8 min, tubes were agitated, and then the contents were allowed to resettle. Next, the supernatant was transferred to the collecting buffer (RPMI with 1% FBS and 2 mM EDTA). After 45-60 min, the tumor tissues were completely dissociated and centrifuged at 4 °C, at 1200 rpm for 5 min. After discarding the supernatant, the Fc block was added to the pellet (1:500) and kept on ice for 15 min. After blocking, tumor cells were stained for flow cytometry by a BD flow cytometer using antibodies against CD45 (1:200), CD4 (1:200), CD8

(1:200), NK1.1 (1:200), Gr1 for MDSC (1:200, for myeloid-derived suppressor cells [MDSC]), CD11b (1:200), Granzyme B (1:100), perforin (1:100)(Supplementary Table 1).

## Single-cell RNA-Sequencing (scRNA-seq)

Human VSs were obtained from *NF2*-SWN patients who provided informed consent, including those who received bevacizumab treatment, in accordance with the Human Subjects protocols approved by the Institutional Review Boards of The Ohio State University and Massachusetts General Hospital. Fresh surgically-removed VS tissues were placed in DMEM or saline and transported to the lab. To prepare single-cell suspension, each tumor was minced and then digested with collagenase I (2 mg/ml) and dispase (1 U/ml), followed by filtering through a 40-μm cell strainer as previously described[62]. Dissociated cells were cryopreserved in 10% dimethyl sulfoxide-containing DMEM plus 10% FBS until processing on the 10X Genomics' Chromium Single Cell platform and sequencing on a NovaSeq 6000 System (Illumina) at a depth of approximately 400 million reads per sample. For computational analysis, we used Cellranger v5.0.1 to align reads to the hg19 human reference sequence. For each sample dataset, unsupervised clustering was performed using the R package Seurat (version 4) (https://www.satijalab.org/seurat; https://www.github.com/satijalab/seurat)[63]. Following normalization using Seurat's NormalizeData function, highly variable genes were identified and used for principal components analysis. Then, we performed clustering using graph-based clustering and visualized using Uniform Manifold Approximation and Projection (UMAP) with Seurat function RunUMAP. To identify the cell type in tumor sample datasets, we input marker gene lists generated by the Seurat FindMarkers function in Toppgene to identify the top cell type makers or cell identities. Differentially expressed genes (DEGs) in a given cell type compared with all other cell types were determined with the FindAllMarkers function in the Seurat package.

## Quantitative RT-PCR

Changes in the mRNA levels were quantified with qPCR, using SYBR Green-based protocol[64]. All qPCR analyses were performed on a Stratagene MX 3000 qPCR System with the operating MXPro qPCR software (Stratagene)[65].

## Western Blot

Thirty micrograms of protein lysates per sample were separated on 10% SDS-polyacrylamide gels[66]. Membranes were probed with antibodies against perforin (1:500) and granzyme B (1:500) (both from Cell Signaling Technology), or β-actin (1:5000, Sigma) as a loading control[67].

## ELISA

Plasma or protein samples obtained from snap-frozen tumors were diluted to a concentration of 2 μg/μl protein levels were measured using mouse multiplex enzyme-linked immunosorbent assay plates following the manufacturer's guidelines (Meso-Scale Discovery). Each sample was tested in triplicate[64,68].

## In vitro cytotoxicity assay

To isolate CD8+ T cells from treated tumors and NK cells from the mouse spleen, tumor tissues and the spleens were placed on a petri dish containing pre-cooled phosphate-buffered saline (PBS) supplemented with 2% FBS. Tumor tissues and the spleens were minced into small pieces and filtered through a 70 μm-pore cell strainer positioned above a 50-ml conical tube (Thermo Fisher Scientific). The petri dish and the strainer were rinsed with pre-cooled PBS supplemented with 2% FBS. Erythrocytes were lysed with ammonium chloride potassium lysing buffer. Subsequently, single cell suspensions were transferred through another 70 μm-pore cell strainer into a new 50-ml conical tube. Cell numbers were counted by trypan blue staining. CD8+ T cells were purified using a MojoSort Mouse CD8 T cell isolation kit (BioLegend), and NK cell fractions were harvested using the NK Cell Isolation Kits and MACS columns (Miltenyi Biotech) following the manufacturer's instructions. Experiments were done when the purity of CD8+ T cells and NK cells was above 90%, as determined by flow cytometry. Cultured mouse lymphoma YAC-1 cells or schwannoma tumor cells were used as the target cells and labeled with calcein AM (10 nM for $1 \times 10^6$ cells/ml, Becton Dickson). After overnight incubation, $1 \times 10^4$ cells/well were seeded in a U-bottom 96-well plate (100 μl/well), protected from light. The effector cells were added in triplicate at the effector:target (E:T) cell ratio of 10:1. Target cells only were plated for baseline fluorescent reading, and target cells in lysis buffer (containing 1% NP40) were included as maximum cell lysis. After incubation for 6 hours, the supernatants were collected, and fluorescent intensity was measured using a fluorescent plate reader. Cytolysis was assessed by measuring the release of calcein AM from target cells and calculated as: effector cell cytotoxicity (%) $= \frac{Sample\ OD - Baseline\ OD}{Maximum\ OD - Baseline\ OD} \times 100\%$ as described previously[69].

## Histological staining

To evaluate tumor cell proliferation and apoptosis, sections of tumor tissues were stained with proliferating cell nuclear antigen (anti-PCNA, 1:1000; Abcam) and TUNEL (ApopTag®; Millipore). Microvessel density (MVD, detected by anti-CD31, 1:200; Millipore) and myofibroblast infiltration (detected by anti-α-smooth muscle actin-Cy3™, 1:200; Sigma-Aldrich) were determined in frozen sections (8 μm). To evaluate the percentage of perfused vessels, 50 μl of FITC-lectin (2 mg/kg) was injected *i.v.* ten minutes before sacrifice to identify perfused vessels. Then, tumors were harvested and embedded in frozen sections (20 μm). Endothelial cell labeling with FITC-lectin demonstrates perfused blood vessels, and CD31+ staining displays all blood vessels in solid tumors. The percentage of FITC-lectin+/CD31+ vessels was quantified using Image J software. To fluorescently label the αPD1 antibody, Antibody Labeling Kit – [FITC (Fluorescein) F10240; ThermoFisher Scientific] was used following the manufacturer's protocol. The labeled αPD1 antibody was processed through a purification column to remove the unbound dye. Four specimens of normal peripheral nerves were obtained postmortem and used as controls. Appropriate positive and negative controls were used for all stains[68,70]. Acquisition was performed on an Olympus IX81 confocal microscope with Fluoviewer FV10-ASW4.2 software. FITC channel shown in magenta in the figures. Histological evaluation using digital quantitative image analysis was performed using ImageJ. Positive staining in 20 random fields/slides was quantified via automated built-in functions based on fluorescent pixel intensity after establishing a threshold to exclude background staining.

## Statistical analyses

Differences in sciatic nerve tumor growth were analyzed using repeated measures two-way ANOVA. Survival curves were generated using the Kaplan-Meier method. Kaplan-Meier survival curves were analyzed by the Log-rank (Mantel-Cox) test. ABR thresholds were analyzed with a linear mixed-effects model. Flow cytometry, ELISA, qPCR, and cytotoxicity studies were analyzed using Student's *t* test and Mann-Whitney U test, as appropriate. All statistical analyses were carried out using GraphPad Prism Software version 9.

## Reporting summary

Further information on research design is available in the Nature Portfolio Reporting Summary linked to this article.

## Ethics approval

The study was approved by MGH's Institutional Animal Care and Use Committee (Protocol #2016N00004).

## Data availability

The raw data generated in this study are provided in the Supplementary Information/Source Data file. The RNASeq data generated in this study have been deposited in the NCBI Gene Expression Omnibus (GEO) database under accession code GSE315579. All data and the supplementary materials from this study are included in this manuscript and are available after publication upon request from the corresponding author. Source data are provided with this paper.

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

## Acknowledgements

We thank Mark Duquette and Anna Khachatryan for their superb technical support, and Dr. Peigen Huang for assisting in animal studies. This study was supported by the NIH RO1-NS126187 and RO1-DC020724 (to L.X.), Department of Defense New Investigator Award (W81XWH-16-1-0219, to L.X.), Investigator-Initiated Research Award (W81XWH-20-1-0222, to L.X.), Clinical Trial Award (W81XWH2210439, to S.R.P. and L.X.), Children's Tumor Foundation Drug Discovery Initiative (to L.X.), Children's Tumor Foundation Clinical Research Award (to L.X. and S.R.P.), and American Cancer Society Mission Boost Award (MBGII-24-1255260-01-MBG to L.X.), CancerFree KIDS (to L.S.C.), and Rally Foundation (L.S.C).

## Author contributions

L.X. and L.S.C. designed the research and supervised the research; S.L., Z.Y., L.W., and J.C. performed mouse model studies; S.L., D.C.B., L.D.L., and R.S. conducted hearing test, L.M.N.W., J.L.O., and L.S.C. conducted scRNA-seq; Z.Y., Y.S., B.X., and A.P.J., performed flow cytometry and histology analysis; W.H. analyzed RNASeq data; L.X., S.L., Z.Y., Y.S., and A.M., analyzed data; L.X., S.R.P., L.S.C., and K.S. wrote the paper.

## Competing interests

The authors declare no competing interests.
