## [Transparent Peer Review file · Nature Communications]

NKG2D Upregulation Sensitizes Tumors to Combined Anti-PD1 and Anti-VEGF Therapy and Prevents Hearing Loss

Corresponding Author: Dr Lei Xu

Version 0:

Reviewer comments:

Reviewer #1

(Remarks to the Author)

The study highlights the synergistic effects of combined α VEGF and α PD1 treatment in an in vivo schwannoma model. The authors demonstrate that α VEGF enhances the anti-tumour efficacy of α PD1, and the combination therapy significantly delayed tumour growth, reduced the number of proliferating cells, and increased apoptotic cell numbers, leading to improved survival in mice compared to monotherapies. The mechanisms underlying α VEGF's action are elucidated, including tumour vasculature normalisation, which enhances α PD1 delivery. Additionally, the α VEGF treatment promotes increased infiltration and activation of cytotoxic T cells and NK cells via upregulation of Natural Killer Group 2 Member D (NKG2D). Importantly the authors revealed that α PD1 treatment in combination with α VEGF, decrease tumour growth of tumours that did not respond to α VEGF.

These findings are a novel contribution to schwannoma research and hold promising potential for future clinical trials involving NF2 patients. α VEGF (bevacizumab) is currently used as a treatment for vestibular schwannomas in NF2 patients, demonstrating partial effectiveness in some patients. Furthermore, several FDA-approved α PD1 inhibitors, which have been previously tested for other conditions, could be repurposed for use in combination therapy with α VEGF in NF2 patients.

Similar co-treatment studies have been conducted in lung carcinoma mouse models, yielding comparable outcomes (doi/full/10.2217/imt-2021-0196). Several clinical trials testing α VEGF and α PD1 combination treatment are currently ongoing. Toxicity reports should be thoroughly evaluated before initiating any clinical trials involving NF2 patients.

The results are consistent with the conclusions and claims; however, there are several aspects that require further attention: Figure 1. Title: B20 should be referred to α VEGF. This also should be addressed throughout the manuscript.

1G: No DAPI staining is visible. Could you clarify which type of microscope was used in the study? The manuscript does not provide any details regarding the microscopy techniques employed.

1I: Flow cytometry data should be presented using histograms or scatter plots, at least a few representative examples from the selected datasets.

3A and G: Same comment as in 1I.

5A-C: Was the effect seen in mice where the switch from α VEGF to α PD1 was done, due to α VEGF pre-treatment?

5E: Same comment as in 1I.

Figure S2. What do the authors mean by 'enhance tumour control'? This statement should be more specific.

Pictures in Figure S2 A are PCNA staining not TUNEL.

Pictures in Figure S2 B are TUNEL staining not PCNA.

Image quantifications: Are these really flow cytometry data?

The legends for all figures should be more detailed and informative.

Additional questions:

1. Will tumours regrow when the treatment is discontinued? This would be important to investigate.
2. Were any adverse events observed in the mice?

Reviewer #2

(Remarks to the Author)

This manuscript claims that combined treatment with anti-PD1 significantly improves the therapeutic effects of anti-VEGF on the prevention of hearing loss as well as tumor growth in their unique model of vestibular schwannomas (VSs) linked to NF2-related schwannomatosis (NF2-SWN). As a mechanism of action, they claims that anti-VEGF normalizes tumor vasculature for drug delivery and immune cell infiltration, and by activating antitumor cytotoxicity of T and NK cells via NKG2D upregulation, leading to enhanced anti-PD1 efficacy and tumor controls. Overall, the topic is of interest in the context of newly suggested combination of immune checkpoint inhibitors such as anti-PD1 with anti-VEGF as a promising therapeutic strategy to treat VS with currently no FDA-approved medications. However, the conclusion of this manuscript is not sufficiently supported by the results provided and requires more experimentation to support the mechanism of action claimed and the rationale of combined anti-VEGF and anti-PD1 therapy in VS.

Major comments

To assess the anti-PD1 drug delivery, the authors fluorescently labeled anti-PD1 antibody with Alexa Fluor 647 labeling kit in histological staining section of Methods (p. 8). This raises the question if the labeled antibody was functionally intact for the binding and blocking of PD1 receptor on target cells. In addition, as they described the labeling of anti-PD antibody with a green fluorescent tag in the Results section (p. 10) and FITC-labeled anti-PD1 antibody (green) in the Figure 1 legend (p. 21), I am wondering if they independently tested two different labeling of anti-PD1 otherwise it is a simple mistake.

p. 12: To profile the T cells and NK cells in VSs with or without bevacizumab treatment, the authors performed scRNA-seq analysis by employing a single VS case treated with bevacizumab for 10 years in comparison to three VS cases without the treatment. Given the heterogeneity of patient response to certain biologics, the authors should include more VS cases in their analysis to draw more convincing conclusions.

p. 13: The authors argued that CD8+ T cells and NK cells are in a suppressed functional state in the VS tumor microenvironment simply based on the results from scRNA-seq analysis for NKG2A, CD94, HLA-E. This should be supported by the functional assessment of isolated CD8+ T cells or NK cells from VS tissues with or without bevacizumab treatment.

p. 15: The approval of bevacizumab in UK along with its therapeutic benefit and limitation is repeatedly described in the Introduction and Discussion, which should be removed and replaced with more insightful discussion based on additional mechanistic studies.

For statistical analyses, they simply used Student's t-test or Mann-Whitney test even for the comparison of multiple groups, time-dependent curves between two groups (may be better with ANOVA), or Kaplan-Meier survival analysis (may be better with log-rank test). In this regard, they should consult the statistician in the Institute and provide the rationale for each analysis. In addition, they should indicate which statistical analysis was used for specific data in the legends along with the presentation of exact p-value instead of $P < 0.01$.

They authors used simple bar graphs for the presentation of the results performed with multiple mice, which should be presented with each dot indicating individual mice to strengthen their results.

For the presentation of flow cytometry analysis of various immune cells, the authors only used bar graphs for each immune cell type in Figure 1I, 3A, 3G, and 5E. In this regard, they are required to provide representative figures and gating strategy in the main figure or supplementary figures.

In in vivo experiments, they commented the use of both male and female mice (1:1 ratio) to ensure sufficient statistical power and to examine any potential sex-bias. Given their use of 6 or 8 mice in each experiment, I am wondering if they used adequate numbers of mice.

In figure 3C and 3D, to claim the activation of CD8+ T cells and NK cells, they performed Western blot analysis for Perforin and Granzyme B proteins and qRT-PCR analysis for mRNA expression of NKG2D and its ligands using tumor lysates. To support their claim, they are required to perform flow cytometric analysis for assessing proteins levels and scRNA-seq for gene expression in a per-cell basis. Moreover, given their claim of NKG2D upregulation as a major mechanism of action, they are required to test the functional effect of NKG2D upregulation such as blockade of NKG2D receptor.

In figure 3E, it seems curious that anti-VEGF treatment enhanced the cytotoxicity of NK cells in the spleen compared to that of CD8+ T cells in the tumor tissues. In this regard, I am wondering if anti-VEGF treatment had a systemic effect on the cytotoxicity of NK cells and CD8+ T cells.

In figure 4 of scRNA-seq analysis, the authors presented the limited set of receptors (NKG2A, CD94, HLA-E) and molecules (GZMB, CCL4, REL, RELB, BFKB1/2). In this regard, they are required to provide more comprehensive data linked to the activation of CD8+ T cells and NK cells (e.g., more receptors including activation markers, immune checkpoint receptors, and etc.)

Minor comments

Provide detailed information on Materials and Methods

- Preparation of tumor lysates for flow cytometry (p. 7), dissociated cells for scRNA-seq (p. 7), and CD8+ T cell isolation from treated tumors for in vitro cytotoxicity assay (p. 8).

Provide detailed information (clone, catalogue number, etc) of antibodies including anti-PD1 and isotype IgG control, perforin and granzyme B

Correct the following errors or typos

p. 8: "Diluted to a concentration of 2 ug/ul inflammatory cytokine levels were measured using" in ELISA section

p. 15: "in preventing preserving hearing...."

p. 16: "This This study...." In Funding section

In figure 1 legend, the statement of "Representative images of In:" appears to cover the data (F), (G), and (H). If the case, the authors are required to indicated the caption (F-H) in front of the statement.

Reviewer #3

(Remarks to the Author)

In their manuscript, Lu et al. demonstrate that combining aPD1 and aVEGF therapy enhances anti-tumor efficacy in syngeneic non-malignant vestibular schwannoma (VS) models and prevents hearing loss. In more detail, aVEGF-induced normalization of tumor blood vessels improved delivery of aPD1 antibodies in preclinical models and is associated with an increased tumor infiltration of effector immune cells in murine and patient VS tissue. Combined aVEGF/aPD1 treatment was effective also after failure of aVEGF monotherapy.

By preclinical development of an effective treatment regimen for non-malignant VS which takes into account not only tumor growth but also hearing function, the authors address a timely topic in the field of neurooncology and may impact future VS management. The study is very well presented and clearly written. The results are supported by a broad methodology, and the inclusion of human post-treatment tissue even strengthens the impact of the study in terms of clinical translation.

Minor comments:

- Please depict individual values and means (+-SEM) in all bar plots.
- Please comment on tolerability and potential side effects beyond ototoxicity.

Version 2:

Reviewer comments:

Reviewer #1

(Remarks to the Author)

The authors have adequately addressed all my concerns and performed the additional experiments requested. After reviewing their responses, including those to other reviewers, I have no outstanding issues. I am happy to recommend this manuscript for publication. The work is novel, relevant to vestibular schwannoma treatment, and shows potential for clinical application.

The statistical evaluation has been corrected using appropriate analytical methods. The methodology is sound, and the data interpretation accurately reflects the results.

Reviewer #2

(Remarks to the Author)

The authors addressed my concerns to a reasonable extent although some key issues related to the inclusion of more VS cases and their functional assessment was discussed as the limitation of the study. Given a rarity of VS case treated long-term with bevacizumab but a conceptual advancement of VS treatment with the combination of anti-VEGF with immune checkpoint inhibitors such as anti-PD1, I recommend the publication of the study in Nature Communications.

Reviewer #3

(Remarks to the Author)

The authors have sufficiently addressed the previous points raised. I believe that the manuscript in its current form is an important contribution to the brain tumor literature and recommend accepting the paper for publication.

NKG2D Upregulation Enhances T and NK Cell Cytotoxicity, Sensitizes Tumors to Combined α PD1 and α VEGF Therapy, and Contributes to Hearing Loss Prevention in Vestibular Schwannoma Model

Corresponding Author: Lei Xu

We thank the reviewers for their helpful comments. We have revised our manuscript to address all of their comments. Our responses are provided below in blue, and the changes made in the revised manuscript are shown in red.

Reviewer #1

The study highlights the synergistic effects of combined α VEGF and α PD1 treatment in an in vivo schwannoma model. The authors demonstrate that α VEGF enhances the anti-tumour efficacy of α PD1, and the combination therapy significantly delayed tumour growth, reduced the number of proliferating cells, and increased apoptotic cell numbers, leading to improved survival in mice compared to monotherapies. The mechanisms underlying α VEGF's action are elucidated, including tumour vasculature normalisation, which enhances α PD1 delivery. Additionally, the α VEGF treatment promotes increased infiltration and activation of cytotoxic T cells and NK cells via upregulation of Natural Killer Group 2 Member D (NKG2D). Importantly the authors revealed that α PD1 treatment in combination with α VEGF, decrease tumour growth of tumours that did not respond to α VEGF.

These findings are a novel contribution to schwannoma research and hold promising potential for future clinical trials involving NF2 patients. α VEGF (bevacizumab) is currently used as a treatment for vestibular schwannomas in NF2 patients, demonstrating partial effectiveness in some patients. Furthermore, several FDA-approved α PD1 inhibitors, which have been previously tested for other conditions, could be repurposed for use in combination therapy with α VEGF in NF2 patients. Similar co-treatment studies have been conducted in lung carcinoma mouse models, yielding comparable outcomes (doi/full/10.2217/imt-2021-0196). Several clinical trials testing α VEGF and α PD1 combination treatment are currently ongoing. Toxicity reports should be thoroughly evaluated before initiating any clinical trials involving NF2 patients. The results are consistent with the conclusions and claims; however, there are several aspects that require further attention.

Response summary: We thank the reviewer for the thorough and positive review of our work. We appreciate the recognition of the novelty and translational potential of our findings. In response to the reviewer's comments, we have:

- i) Addressed all points in detail,
- ii) Performed new experiment investigating whether tumors grow after discontinuation of therapy (**Supplemental Fig. 2A**),
- iii) Performed new experiment assessing body weight loss and tolerability (**Supplemental Fig. 2C-2D**).
- iv) Added references to α VEGF and α PD1 combination treatment preclinical studies and clinical trials,

1. Figure 1. Title: B20 should be referred to α VEGF. This also should be addressed throughout the manuscript.

Reply: As suggested by the reviewer, we have replaced all 'B20' with ' α VEGF' throughout the manuscript, including the main text, figures, and figure legends.

2. 1G: No DAPI staining is visible. Could you clarify which type of microscope was used in the study? The manuscript does not provide any details regarding the microscopy techniques employed.

Reply: We apologize for the lack of clarity in the previous version. The image shown in Figure 1H (previously Fig 1G) was acquired using confocal microscopy. In this experiment, we visualized the distribution of anti-PD1 antibody conjugated to a green fluorescent tag to assess antibody delivery. To better visualize the green fluorescence signal, we omitted the DAPI channel, as the strong nuclear staining could obscure the green signal. To address the reviewer's concern, we have now added a detailed description of confocal microscopy methods in the Materials and Methods section (p9).

3. 1I: Flow cytometry data should be presented using histograms or scatter plots, at least a few representative examples from the selected datasets.

Reply: We appreciate the reviewer's helpful suggestion. Following the recommendation, we have replaced Fig. 1I flow cytometry data with scatter plots in the revised **Figure 1J**, and we included representative gating strategies in **Supplemental Figure 1** and **Supplemental Figure 4**.

Figure 1J. Flow cytometry quantification of CD8⁺ T cell, NK cells and CD8/T_{reg} ratio in *Nf2^{-/-}* tumors treated with control-IgG and αVEGF.

4. 3A: Same comment as in 1I.

Reply: As suggested by the reviewer, we have replaced the Fig. 3A flow cytometry data with scatter plots in the revised **Figure 3A**.

Figure 3A. Flow cytometry analysis of the number of CD8⁺ T cell, NK cell and MDSC in *Nf2^{-/-}* tumors treated with control IgG (Ctrl), αPD1, αVEGF, and combination antibodies (Comb). *P<0.05.

5. 5A-C: Was the effect seen in mice where the switch from α VEGF to α PD1 was done, due to α VEGF pre-treatment?

Reply: We thank the reviewer for this insightful question. The enhanced tumor control observed after switching from α VEGF to α PD1 likely reflects a sustained effect of α VEGF-induced vascular normalization. As shown in Figure 1C, α VEGF treatment increases pericyte coverage, making tumor vessels structurally similar to normal vasculature. Correspondingly, Figure 1D-E demonstrated that this structural normalization leads to normalized vessel perfusion, which facilitates intratumoral immune cell infiltration (Figure 1J), thereby creating a more immunostimulatory microenvironment that can potentiate α PD1-mediated cytotoxicity. We have added a discussion of this temporal relationship between α VEGF pre-treatment and α PD1 efficacy in the Discussion section (p16).

6. 5E: Same comment as in 1I.

Reply: As suggested by the reviewer, we have replaced the old Fig. 5E flow cytometry data with scatter plots in the revised **Figure 5E**.

Figure 5E. Flow cytometry analysis of the number of CD8⁺ T cell, NK cell, and MDSC in *Nf2*^{-/-} tumors from different treatment groups.

7. Figure S2. What do the authors mean by 'enhance tumour control'? This statement should be more specific.

Reply: We have revised the title of Figure S3 (previously Figure S2) to read: 'Combined α VEGF and α PD1 treatment reduces proliferating and increases apoptotic tumor cells compared to control and monotherapies in *Nf2*^{-/-} model'.

8. Pictures in Figure S2 A are PCNA staining not TUNEL. Pictures in Figure S2 B are TUNEL staining not PCNA. Image quantifications: Are these really flow cytometry data?

Reply: We apologize for the labeling error. The text has been corrected to indicate that Figure S3A (previously Figure S2A) shows PCNA staining, and Figure S3B shows TUNEL staining. We also clarified in the figure legend that the quantification data were obtained from image analysis, not flow cytometry. See Result section p11.

9. The legends for all figures should be more detailed and informative.

Reply: We have revised all figure legends to provide more detailed information about the experiments.

Additional questions:

1. Will tumours regrow when the treatment is discontinued? This would be important to investigate.

Reply: As suggested by the reviewer, we conducted a new experiment to address this question. In a separate cohort of *Nf2*^{-/-} sciatic nerve tumor-bearing mice, all treatments were discontinued after three doses.

Remarkably, tumor growth remained suppressed even after treatment cessation, indicating a durable antitumor effect that persisted beyond the active treatment phase. Please see Result (p11) and **Supplemental Figure 2C-D**.

New animal experiment

Supplemental Figure 2. (C). Diagram showing the timeline of α VEGF and α PD1 combination treatment in mice bearing *Nf2*^{-/-} tumors in the sciatic nerve. We discontinued all treatments after administering three doses. **(D)** Tumor diameter was measured by caliper every 3 days post-treatment.

2. Were any adverse events observed in the mice?

Reply: In both sciatic nerve and CPA models, all treatments - including monotherapies and the combination therapy - were well tolerated, with no significant body weight loss observed in any treatment group. This new data has now been added to Result (p10-11) and **Supplemental Figure 2A**. We also showed that α PD1 does not affect hearing function in non-tumor bearing mice, indicating that α PD1 does not cause acute ototoxicity in this mouse model. These results were included in **Supplemental Figure 2E-F**.

New animal data

Supplemental Figure 2A. Bodyweight of mice was measured at the indicated timepoints and expressed as a percentage of the initial body weight before tumor implantation on day 0. No significant body weight loss (defined as >15% change) was observed in any treatment group.

Reviewer #2:

This manuscript claims that combined treatment with anti-PD1 significantly improves the therapeutic effects of anti-VEGF on the prevention of hearing loss as well as tumor growth in their unique model of vestibular schwannomas (VSs) linked to NF2-related schwannomatosis (NF2-SWN). As a mechanism of action, they claims that anti-VEGF normalizes tumor vasculature for drug delivery and immune cell infiltration, and by activating antitumor cytotoxicity of T and NK cells via NKG2D upregulation, leading to enhanced anti-PD1 efficacy and tumor controls. Overall, the topic is of interest in the context of newly suggested combination of immune checkpoint inhibitors such as anti-PD1 with anti-VEGF as a promising therapeutic strategy to treat VS with currently no FDA-approved medications. However, the conclusion of this manuscript is not sufficiently supported by the results provided and requires more experimentation to support the mechanism

of action claimed and the rationale of combined anti-VEGF and anti-PD1 therapy in VS.

Response summary: We thank the reviewer for their thoughtful and encouraging comments regarding the significance of our study and the potential of combining α PD1 with α VEGF as a therapeutic strategy for vestibular schwannomas. We appreciate the reviewer's suggestion that additional data are needed to further substantiate the proposed mechanism of action. In response to the reviewer's comments, we have:

- i) Addressed all points in detail,
- ii) Performed new flow cytometry experiment assessing CD8 T cell and NK cell activation (**Figure 3C**).
- iii) Performed new animal experiment using NKG2D blockade to investigate if α VEGF enhances α PD1 efficacy via promoting NKG2D-mediated cytotoxic (**Figures 3I-3J**).
- iv) Expanded scRNASeq analysis to profile T cell and NK cell activation landscape (**Figure 4 and Supplemental Figure 5**)

Major comment:

1. To assess the anti-PD1 drug delivery, the authors fluorescently labeled anti-PD1 antibody with Alexa Fluor 647 labeling kit in histological staining section of Methods (p. 8). This raises the question if the labeled antibody was functionally intact for the binding and blocking of PD1 receptor on target cells.

Reply: We thank the reviewer for this important point. Our earlier description may not have been sufficiently clear. The fluorescently labeled α PD1 antibody was not used for treatment, it was used exclusively to assess antibody delivery following α VEGF-induced vascular normalization.

In this experiment, mice bearing $Nf2^{-/-}$ schwannomas in the CPA were treated with either control IgG or α VEGF for three weeks. Then, FITC-conjugated α PD1 was administered intraperitoneally. Twenty-four hours later, tumor-bearing brains were collected and for vascular staining and confocal imaging to evaluate the distribution of the fluorescent α PD1 antibody.

To improve clarity, we have revised the Results section (p10) and added a schematic illustration in the new **Figure 1F** to clarify this experimental design.

Figure 1F. We treated mice bearing $Nf2^{-/-}$ schwannomas in the CPA with either control IgG- or α VEGF for three weeks. FITC-conjugated α PD1 antibody (green) were injected *i.p.*. Twenty-four hours later, tumor-bearing brains were harvested for vascular staining, and imaged by confocal microscopy to visualize the distribution of fluorescently labeled α PD1 antibody.

In addition, as they described the labeling of anti-PD antibody with a green fluorescent tag in the Results section (p. 10) and FITC-labeled anti-PD1 antibody (green) in the Figure 1 legend (p. 21), I am wondering if they independently tested two different labeling of anti-PD1 otherwise it is a simple mistake.

Reply: We have corrected the text to ensure a consistent description of the fluorescent labeling throughout the manuscript. The α PD1 antibody was labeled with FITC (Fluorescein), which emits green fluorescence (Antibody Labeling Kit, Cat#F10240, Thermo Fisher Scientific). This information has been added to the Methods section (p9).

2. p. 12: To profile the T cells and NK cells in VSs with or without bevacizumab treatment, the authors performed scRNA-seq analysis by employing a single VS case treated with bevacizumab for 10 years in comparison to three VS cases without the treatment. Given the heterogeneity of patient response to certain biologics, the authors should include more VS cases in their analysis to draw more convincing conclusions.

Reply: We agree with the reviewer that additional VS cases would strengthen the analysis and better represent patient heterogeneity in response to bevacizumab. However, VS is a rare disease, and surgical samples from patients treated long-term with bevacizumab are extremely limited, as surgery is generally avoided whenever possible due to the substantial risks of further nerve damage, causing hearing loss, facial paralysis, and chronic dizziness. We hope the reviewer agrees with us that obtaining more VS cases is challenging. We have now acknowledged this limitation and emphasized the importance of expanding the scRNA-seq analysis in future studies as more patient samples become available, please see Discussion section (p17).

3. p. 13: The authors argued that CD8⁺ T cells and NK cells are in a suppressed functional state in the VS tumor microenvironment simply based on the results from scRNA-seq analysis for NKG2A, CD94, HLA-E. This should be supported by the functional assessment of isolated CD8⁺ T cells or NK cells from VS tissues with or without bevacizumab treatment.

Reply: We agree with the reviewer that scRNASeq data alone cannot provide direct functional evidence of CD8⁺ T cell and NK cell activity. To address this point, we evaluated the functional status of CD8 T cells and NK cells in our mouse model with and without anti-VEGF treatment (**Figures 3C, 3G**), which supports our conclusions regarding their activation state.

We fully recognize that direct functional assays on human VS-infiltrating lymphocytes would provide the most definitive evidence. However, VS is a rare disease, and isolating sufficient numbers of patient CD8 T cells and NK cells is highly challenging. To avoid overinterpretation of our human data, we have removed the original statement regarding patient CD8 T cell and NK cell function from the Results section and revised the text to more accurately reflect these limitations.

4. p. 15: The approval of bevacizumab in UK along with its therapeutic benefit and limitation is repeatedly described in the Introduction and Discussion, which should be removed and replaced with more insightful discussion based on additional mechanistic studies.

Reply: As suggested by the reviewer, we have removed the redundant discussion of bevacizumab's approval and therapeutic limitations from the Introduction and revised the Discussion to focus on the mechanistic insights gained from our study and their potential clinical implications (page 16-17).

5. For statistical analyses, they simply used Student's t-test or Mann-Whitney test even for the comparison of multiple groups, time-dependent curves between two groups (may be better with ANOVA), or Kaplan-Meier survival analysis (may be better with log-rank test). In this regard, they should consult the statistician in the Institute and provide the rationale for each analysis. In addition, they should indicate which statistical analysis was used for specific data in the legends along with the presentation of exact p-value instead of $P < 0.01$.

Reply: As suggested by the reviewer, we consulted our biostatistician, Dr. Alona Muzikansky (a co-author on the manuscript), and re-performed all statistical analyses accordingly. We have specified the statistical methods used for each dataset in the corresponding figure legends and reported exact p-values.

6. They authors used simple bar graphs for the presentation of the results performed with multiple mice, which should be presented with each dot indicating individual mice to strengthen their results.

Reply: As suggested by the reviewer, we have revised all figures with bar graphs and replaced them with new dot blots, displaying individual data points for each mouse to better illustrate data distribution. Please see the

new **Figures 1J, 3A, 3C, 3E** and **5E** in our response to Reviewer 1 above (Comments #3, 4, and 6), and the new **Figures 1B, 1C, 1E, 1I,** and **3B** below.

Figure 1 and Figure 3. New dot blots.

7. For the presentation of flow cytometry analysis of various immune cells, the authors only used bar graphs for each immune cell type in Figure 1I, 3A, 3G, and 5E. In this regard, they are required to provide representative figures and gating strategy in the main figure or supplementary figures.

Reply: As suggested by the reviewer, we have updated the flow cytometry data with dot plots in the revised **Figures 1, 3,** and **5** (please see revised graphs in our response to Reviewer 1, and Comments #6 above). We included representative gating strategies in **Supplemental Figure 1** and **Supplemental Figure 4**.

Figure S1. Gating strategy for CD8⁺ T cells and NK cells in flow cytometry analysis.

Figure S4. Gating strategy for Granzyme B and perforin in CD8⁺ T cells and NK cells in flow cytometry analysis.

8. In in vivo experiments, they commented the use of both male and female mice (1:1 ratio) to ensure sufficient statistical power and to examine any potential sex-bias. Given their use of 6 or 8 mice in each experiment, I am wondering if they used adequate numbers of mice.

Reply: We appreciate the reviewer's comment. All in vivo data shown in the figures are representative of at least three independent experiments. We have updated the N values in the figure legends to accurately reflect the total number of mice used across all experiments, which demonstrates that the sample sizes were adequate for the analyses performed.

9. In figure 3C and 3D, to claim the activation of CD8⁺ T cells and NK cells, they performed Western blot analysis for Perforin and Granzyme B proteins and qRT-PCR analysis for mRNA expression of NKG2D and its ligands using tumor lysates. To support their claim, they are required to perform flow cytometric analysis for assessing proteins levels and scRNA-seq for gene expression in a per-cell basis.

Reply: As suggested by the reviewer, we performed new flow cytometry analyses to assess Granzyme B and Perforin expression in CD8⁺ T cells and NK cells. The previous Western blot data have been replaced with these new flow cytometry results, now presented in the revised **Figure 3C**, p28.

Figure 3C. Flow cytometry analysis of the proportion of CD8⁺ T cells and NK cells expressing granzyme B and perforin in *Nf2*^{-/-} tumors treated with control IgG or αVEGF antibodies.

10. Moreover, given their claim of NKG2D upregulation as a major mechanism of action, they are required to test the functional effect of NKG2D upregulation such as blockade of NKG2D receptor.

Reply: Following the reviewer's suggestion, we performed additional experiments using an NKG2D neutralizing antibody to evaluate the functional role of NKG2D signaling in mediating the therapeutic benefit of the combination therapy.

Our new data showed that blockade of NKG2D markedly attenuates the antitumor effects of both anti-VEGF monotherapy and the combination treatment, confirming the critical role of NKG2D in this mechanism. These results are now included in the revised **Result section** (p12) and in the new **Figure 3I-3J** (p28).

Figure 3I-3J. (A) Schematic and timeline of NKG2D blockade in *Nf2*^{-/-} sciatic nerve model. Mice bearing *Nf2*^{-/-} tumors in the sciatic nerve were treated with anti-NKG2D (αNKG2D, 200 μg/mouse, *i.p.*, for two doses: seven days before implantation and seven days after implantation), followed by control IgG, αPD1, αVEGF antibody treatments. (B) Sciatic nerve *Nf2*^{-/-} tumor diameter was measured by caliper every three days following treatment. αVEGF vs. αVEGF+αNKGD2: P=0.02 (red); αVEGF+αPD1 vs. αVEGF+αPD1+αNKGD2: P=0.04 (blue).

11. In figure 3E, it seems curious that anti-VEGF treatment enhanced the cytotoxicity of NK cells in the spleen compared to that of CD8⁺ T cells in the tumor tissues. In this regard, I am wondering if anti-VEGF treatment had a systemic effect on the cytotoxicity of NK cells and CD8⁺ T cells.

Reply: We thank the reviewer for this insightful comment. The enhanced cytotoxicity of NK cells observed in the spleen following anti-VEGF treatment likely reflects a systemic immune-modulatory effect of VEGF blockade. VEGF is known to suppress immune cell maturation and cytotoxic function; thus, its inhibition can restore NK cell activity in peripheral lymphoid organs (64). The activation of tumor-associated CD8⁺ T cells is more dependent on tumor-specific antigen presentation within the tumor microenvironment. Future studies are needed to further investigate the systemic versus local immune effects of anti-VEGF treatment.

We have now added this discussion in the Discussion section (p16).

12. In figure 4 of scRNA-seq analysis, the authors presented the limited set of receptors (NKG2A, CD94, HLA-E) and molecules (GZMB, CCL4, REL, RELB, BFKB1/2). In this regard, they are required to provide more comprehensive data linked to the activation of CD8⁺ T cells and NK cells (e.g., more receptors including activation markers, immune checkpoint receptors, and etc.)

Reply: As suggested by the reviewer, we have expanded our scRNA-seq analysis to comprehensively profile the landscape of T cell and NK cell activation, including co-stimulatory and co-inhibitory receptors, cytokines and chemokines, immune checkpoint molecules, and transcription factors, in naïve and bevacizumab-treated *NF2-SWN* patient samples. These results are presented in the new **Figure 4C** and **Supplementary Figure 5**. The complete scRNA-seq dataset will be deposited in the GEO database upon publication.

Figure 4C. In bevacizumab-treated VS, compared to naïve VS, we observed changes in RNA expression of CD8⁺ T cell and NK cell cytotoxicity markers: i) reduced expression of inhibitory receptors on NK cells and T cells, including *KIR2DL3*, *KIR2DL2*, *KIR3DL1*, *KIR3DL2*^{44,45}, ii) reduced expression of immune checkpoint proteins and receptors, including *CD96*, *PVRIG*, *HLA-E*^{46,47,48}, iii) increased expression of molecules mediating NK cell and CD8⁺ T cell cytotoxicity and recruitment, including Granzyme B (*GZMB*), the key mediator of⁴⁹, C-C motif chemokine ligand 4 (*CCL4*), chemokine recruiting NK cells and T cells⁵⁰, *CLEC2B* and *SH2D1B* which activates NK cells^{51,52}, and iv) NFκB pathway genes, including *REL*, *NFKB1*, and *NFKB2*, which play a crucial role in regulating the cytotoxic activity of both NK cells and T cells⁵³ (Figures 4D).

New RNASeq analysis - Supplemental Figure 5

Figure S5. T cell and NK cell activation, co-stimulatory, and co-inhibitory receptor landscape in Naïve vs. bevacizumab-treated *NF2-SWN* patient samples. Dot plot showing scaled average expression (color) and percent of cells expressing each gene (dot size) across major T cell and NK cell subsets. Rows represent individual immune cell populations in naïve samples (blue label) and bevacizumab-treated samples (red labels).

Minor comments

- Provide detailed information on Materials and Methods: Preparation of tumor lysates for flow cytometry (p. 7), dissociated cells for scRNA-seq (p. 7), and CD8⁺ T cell isolation from treated tumors for in vitro cytotoxicity assay (p. 8).

Reply: As requested by the reviewer, we have added detailed descriptions of the procedures for tumor lysates preparation for flow cytometry (p7), cell dissociation for scRNA-seq (p7), and CD8⁺ T cell isolation from treated tumors for in vitro cytotoxicity assay (p8).

- Provide detailed information (clone, catalogue number, etc) of antibodies including anti-PD1 and isotype IgG control, perforin and granzyme B

Reply: As requested by the reviewer, we have updated **Supplemental Table 1** to include detailed information on all flow cytometry antibodies and treatment antibodies.

Table S1. Antibody panels.

Flow cytometry antibodies			
Surface Markers	Clone	Fluorophore	Catalog Number
CD11b	M1/70	APC-Cy7	BioLegend 101212
CD3	17A2	PE-cy7	BioLegend 100219
CD4	GK1.5	FITC	BioLegend 100405
CD45	30-F11	PerCP	BioLegend 103130
CD8	53-6.7	APC-cy7	BioLegend 100713
Gri1	RB6-8C5	PE	BioLegend 108408
NK1.1	S17016D	APC	BioLegend 156505
Granzyme B	GB11	FITC	ThermoFisher 12-8898-82
Perforin	eBioOMAK-D	PE/Dazzl 594	ThermoFisher 14-9392-82
Treatment antibodies			
Target	Clone	Company	Catalog Number
Anti-VEGF	B20-4.4	Genentech	Under MTA
Isotype control IgG	2A3	BioXCell	BE0089
Anti-PD1	RMP1-14	BioXCell	BE0146
Anti-CD8	2.43	BioXCell	BE0061
Anti-NK	PK136	BioXCell	BE0036
Anti-NKG2D	CX5	BioXCell	BE0034

Supplemental Table 1. Updated information for antibodies.

- Correct the following errors or typos

Reply: We thank the reviewer for the careful review of our manuscript. All typographical and formatting errors noted have been corrected as detailed in the responses below.

- p. 8: “Diluted to a concentration of 2 ug/ul inflammatory cytokine levels were measured using” in ELISA section

Reply: Corrected the sentence in the ELISA section to read properly (“diluted to a concentration of 2 µg/µl protein levels, p8).

- p. 15: “in preventing preserving hearing....”

Reply: Corrected the phrase to remove redundancy (p15).

- p. 16: “This This study....” In Funding section

Reply: Corrected the duplicated word in the Funding section (p18).

- In figure 1 legend, the statement of “Representative images of In:” appears to cover the data (F), (G), and (H). If the case, the authors are required to indicated the caption (F-H) in front of the statement.

Reply: We revised the Figure legend to clarify that the statement “(F-G) Investigate the effects of αVEGF treatment on αPD1 drug delivery” (p25).

Reviewer #3:

In their manuscript, Lu et al. demonstrate that combining aPD1 and aVEGF therapy enhances anti-tumor efficacy in syngeneic non-malignant vestibular schwannoma (VS) models and prevents hearing loss. In more detail, aVEGF-induced normalization of tumor blood vessels improved delivery of aPD1 antibodies in preclinical models and is associated with an increased tumor infiltration of effector immune cells in murine and patient VS tissue. Combined aVEGF/aPD1 treatment was effective also after failure of aVEGF monotherapy. By preclinical development of an effective treatment regimen for non-malignant VS which takes into account not only tumor growth but also hearing function, the authors address a timely topic in the field of neurooncology and may impact future VS management. The study is very well presented and clearly written. The results are supported by a broad methodology, and the inclusion of human post-treatment tissue even strengthens the impact of the study in terms of clinical translation.

Response summary: We sincerely thank the reviewer for the positive and encouraging comments on the significance and clarity of our study. We appreciate the recognition of our preclinical approach, which integrates both tumor control and hearing preservation, as well as the translational relevance strengthened by the inclusion of human post-treatment VS samples. In response to the reviewer’s comments, we have addressed all points in detail and add new data on tolerability assessment (**Supplemental Fig. 2A**).

Minor comments:

- Please depict individual values and means (+-SEM) in all bar plots.

Reply: As suggested by the reviewer, all bar graphs have been revised to dot blots, displaying individual data points across all Figures.

- Please comment on tolerability and potential side effects beyond ototoxicity.

Reply: In both sciatic nerve and CPA models, all treatments - including monotherapies and the combination therapy - were well tolerated, with no significant body weight loss observed in any treatment group. These data have now been added to the new **Supplemental Figure 2A**.

New animal data

Supplemental Figure 2A. Bodyweight of mice was measured at the indicated timepoints and expressed as a percentage of the initial body weight before tumor implantation on day 0. No significant bodyweight loss (defined as >15% change) was observed in any treatment group.

NKG2D Upregulation Enhances T and NK Cell Cytotoxicity, Sensitizes Tumors to Combined α PD1 and α VEGF Therapy, and Contributes to Hearing Loss Prevention in Vestibular Schwannoma Model

Corresponding Author: Lei Xu

Reviewer #1:

The authors have adequately addressed all my concerns and performed the additional experiments requested. After reviewing their responses, including those to other reviewers, I have no outstanding issues. I am happy to recommend this manuscript for publication. The work is novel, relevant to vestibular schwannoma treatment, and shows potential for clinical application.

The statistical evaluation has been corrected using appropriate analytical methods. The methodology is sound, and the data interpretation accurately reflects the results.

Response:

We sincerely thank the reviewer for their careful re-evaluation of our manuscript and for their positive assessment. We are grateful for the reviewer's recognition of the novelty, methodological rigor, and translational relevance of our work, as well as for acknowledging the additional experiments and corrected statistical analyses. We appreciate the reviewer's support for publication.

Reviewer #2:

The authors addressed my concerns to a reasonable extent although some key issues related to the inclusion of more VS cases and their functional assessment was discussed as the limitation of the study. Given a rarity of VS case treated long-term with bevacizumab but a conceptual advancement of VS treatment with the combination of anti-VEGF with immune checkpoint inhibitors such as anti-PD1, I recommend the publication of the study in *Nature Communications*.

Response:

We thank the reviewer for their thoughtful and constructive evaluation of our study. We appreciate the reviewer's understanding of the limitations imposed by the rarity of vestibular schwannoma cases treated long-term with bevacizumab, and we have explicitly discussed these constraints in the revised manuscript. We are grateful for the reviewer's recognition of the conceptual advancement represented by combining anti-VEGF therapy with immune checkpoint inhibition and for their recommendation to publish our work in *Nature Communications*.

Reviewer #3:

The authors have sufficiently addressed the previous points raised. I believe that the manuscript in its current form is an important contribution to the brain tumor literature and recommend accepting the paper for publication.

Response:

We sincerely thank the reviewer for their careful reassessment of the revised manuscript and for their positive feedback. We are grateful for the reviewer's recognition of the significance of this work and their recommendation for publication.